# Reforestation Will Lead to a Long-Term Downward Trend in the Water Content of the Surface Soil in a Semi-Arid Region

**Junjun Yang** [1],*, **Lingxia Guo** [1], **Yufeng Liu** [1], **Pengfei Lin** [2] and **Jun Du** [2]

1   College of Geography and Environment, Xianyang Normal University, Xianyang 712000, China; guolingxia1984@163.com (L.G.); yfliu518@163.com (Y.L.)
2   Linze Inland River Basin Research Station, Chinese Ecosystem Research Network, Key Laboratory of Eco-Hydrology of Inland River Basin, Northwest Institute of Eco-Environment and Resources, Chinese Academy of Sciences, Lanzhou 730000, China; linpengfei@lzb.ac.cn (P.L.); dujun159@126.com (J.D.)
*   Correspondence: junjun_yang@126.com; Tel.: +86-29-3372-2032

**Abstract:** The spatial distribution of soil moisture is a critical determinant for the success of vegetation restoration initiatives in semi-arid and arid regions. The Qilian Mountains, situated within a semi-arid zone in China, have been subject to significant water-induced soil erosion, which has led to extensive restoration activities, predominantly utilizing the species *P. crassifolia*. However, the interconnections between soil moisture and various land cover types within this region remain unclear, presenting challenges to effective woodland rehabilitation. This study examines the surface soil moisture dynamics in afforested areas with varying ages of plantation to determine the influence of tree planting on the moisture content of the upper soil layer. It investigates the characteristics and temporal patterns of surface soil moisture as the age of the plantation increases. The findings indicate that: (1) soil moisture levels follow a descending sequence from natural forest, through shrubland and grassland, to planted forest and mixed forest, with statistically significant differences observed between natural and mixed forests ($p < 0.05$); (2) young afforested areas (less than 50 years old) have lower soil moisture levels compared to natural forests, shrublands, or grasslands, and the ecohydrological impacts of afforestation become apparent with a temporal delay; and (3) the analysis using Generalized Additive Mixed Models (GAMM) and the application of Kriging interpolation to determine the spatial distribution of soil moisture reveals that in semi-arid and arid regions, several factors have a pronounced a non-linear relationship with the moisture content of the surface soil. These factors include the duration of afforestation, the position on the lower slope, the presence of shade on the slope, and the scale at which the study is conducted. Therefore, a comprehensive understanding of the dynamics of soil water content is essential to prevent the potential failure of artificially established forests due to inadequate soil moisture in their later stages.

**Keywords:** soil moisture; spatial heterogeneity; semi-arid region; plantation; Generalized Additive Mixed Model (GAMM)

## 1. Introduction

The spatial distribution of surface soil moisture is significant for understanding the interaction between vegetation type and soil moisture in semi-arid and arid areas [1–3], revealing how hydrological dynamics are linked to ecological patterns and processes [4–6]. Indeed, the quantity and spatial distribution of soil moisture directly influences the patterns and population abundance of vegetation [7,8]. In semi-arid regions, the presence and absence of seasonal vegetation are highly correlated with soil water content [9], while areas with sparse vegetation are predominantly affected by local factors such as slope and soil texture [3,10].

The Qilian Mountains in northwestern China are characterized by a semi-arid and arid climate. This region is also the source of numerous inland rivers, which support

ecosystem stability in the arid Hexi River Corridor. Several afforestation projects have been implemented to conserve soil and water, control desertification, and produce timber [11]. *P. crassifolia* is the dominant species in the region [12]. Consequently, the planting and management of *P. crassifolia* forests will alter hydrological transport processes and the interaction between moisture and the atmosphere [13,14]. However, the internal mechanisms by which large-scale artificial forests evolve into typical patchy forest landscape patterns are not well understood, presenting an urgent scientific issue that requires in-depth investigation.

The spatial pattern of soil moisture is influenced by various environmental factors, including soil type, topography, vegetation, and the amount of local precipitation [1]. In a loess hilly region of China, soil water content determines a self-organized vegetation pattern in arid environments, whereas, in humid regions, soil water content is not the decisive factor for vegetation patterns [15]. Liu analyzed vegetation patterns and hydraulic properties of surface soils in an alpine catchment, indicating that the surface soil moisture pattern was primarily controlled by terrain-related processes [16]. Geostatistical tools such as kriging interpolation following a semivariogram analysis are employed to characterize and estimate the variability of soil moisture [17]. Ordinary kriging, derived from spatially interpolated soil moisture maps, revealed that the variability of surface soil moisture was considerable, with the distribution exhibiting small patches [18]. A map of interpolation kriging was utilized to examine the relationship between soil moisture distribution and topography [19].

At a small spatial scale, heterogeneity in land use, vegetation type, and soil characteristics are the dominant factors affecting soil moisture dynamics [20,21]. Additionally, soil moisture is contingent upon antecedent precipitation conditions in the study area [12,21,22]. Precipitation serves as the principal determinant of vegetative shifts within arid regions; owing to the high rates of evaporation, a diminution in precipitation levels can precipitate a swift decline in the moisture content of the soil. This reduction in soil hydration has consequential impacts on critical physiological functions, including the transpiration of regional vegetation [22]. Jian posits that in semi-arid regions, the primary mechanisms for plant water absorption are nocturnal root uptake and diurnal evapotranspiration, highlighting the critical role of soil moisture in supporting the physiological functions and development of vegetation in arid environments [23]. Consequently, soil water content is the main limiting factor for land cover and vegetation succession in arid regions, and any alteration in hydrological processes may cause significant ecosystem disturbances and result in new landscape patterns [1,16,24–26].

Forest planting in China has been extensively promoted over the past two decades through the "Grain for Green" projects, which aimed to convert low-quality croplands into forests, shrublands, and grasslands to prevent ecosystem degradation [27,28]. However, investigations conducted decades after the establishment of artificial forests have revealed high rates of tree mortality [27,29]. The amount of soil moisture within a 100 cm depth across all land cover types in the Loess Plateau decreased from 2009 to 2013 [23]. Soil desiccation induced by the proliferation of incompatible plant species is not readily reversible and the reestablishment of vegetation utilizing unsuitable species may precipitate a decline in vegetative health. The alteration of ecosystems in arid zones is characterized by unpredictability and irreversibility, which mandates a prudent strategy when establishing man-made woodlands. Consequently, an in-depth grasp of the factors that govern the spatial distribution of soil moisture is imperative to guarantee the enduring viability of these ecosystems [3].

The survey indicated that grasslands and shrublands are the original vegetative states of the study area. Young plantation forests are indicative of newly established plantations, mixed forests represent an intermediate stage in the succession of plantation forests, and mature forests correspond to the late-stage vegetative states of plantation forests [30,31]. The objective was to enhance the understanding of forest plantations in arid regions by providing empirical support for research on the forest–water nexus within the region. The

specific goals were to (a) quantify the trend of surface soil moisture evolution at different afforestation stages, (b) characterize the spatial heterogeneity of soil moisture in reforested areas of varying ages, and (c) investigate the intrinsic relationship between surface soil moisture content in arid regions and factors such as vegetation type, forest age, vegetation density, slope position, aspect, and gradient.

## 2. Methods

### 2.1. Site Description

The research was conducted within the Dahuang Mountain Forest Reserve (100°22′ E, 38°43′ N, Figure 1), situated in the Qilian Mountains. The reserve is located approximately 45 km southeast of Shandan County, Gansu Province, China, in the northern sector of the Qilian Mountains, which is characterized by an arid climate. The region has been the focus of several afforestation initiatives, providing an optimal setting for investigating the spatial patterns of soil moisture subsequent to vegetative change. The climate is classified as semi-arid and cold temperate, with an average annual temperature of 2.5 °C and mean annual precipitation of 385 mm (recorded from 1994 to 2014), with about 80% of the precipitation occurring during the warmer months from June to September [17]. The soil types are identified as gray cinnamon (according to the FAO classification system) on shaded and semi-shaded slopes and chestnut (as per Chinese soil taxonomy) on sunny slopes [30].

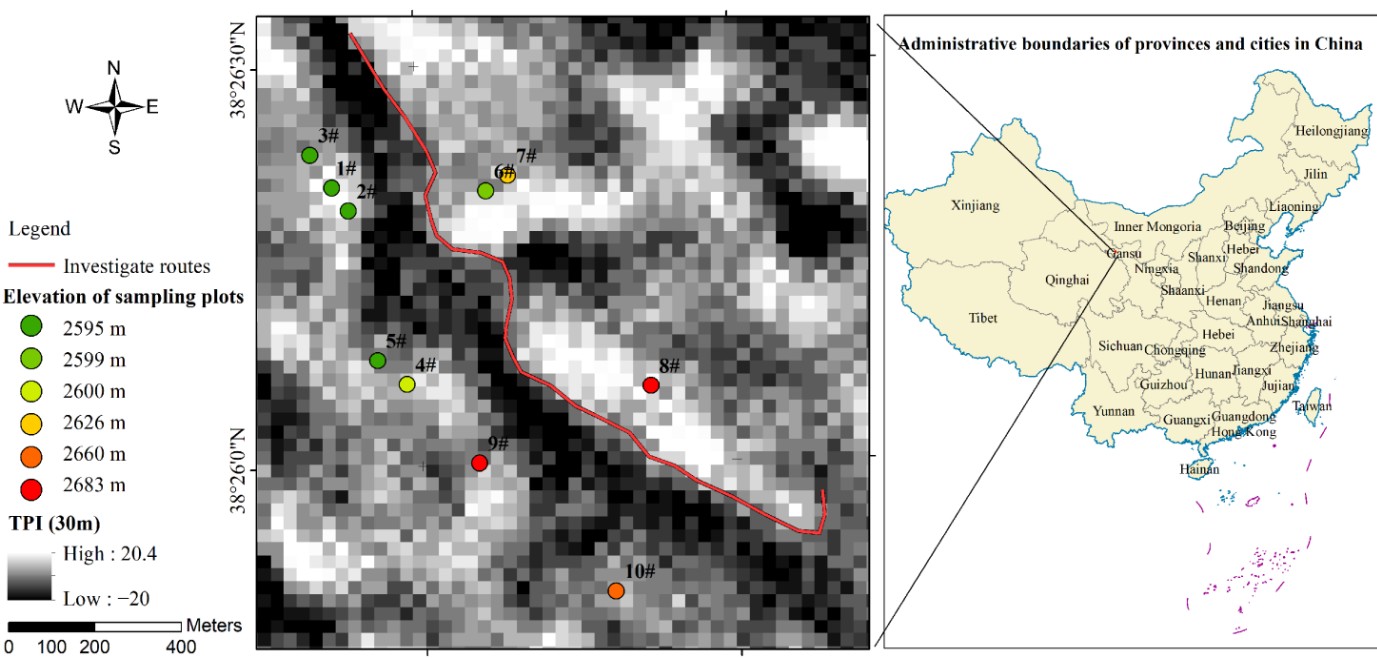

**Figure 1.** Study plots within the Dahuang Mountain of the Qilian Mountains employ the Topographic Position Index (TPI) with a 30 m pixel resolution to characterize topographic positions, such as ridges, canyon bottoms, and slopes. Positive TPI values denote ridgetops and hilltops; negative values signify valleys and canyon bottoms; and values near zero represent flat areas or mid-slope positions [31]. The TPI was derived from the local Digital Elevation Model (DEM) with a resolution of 30 m.

The forests, almost exclusively composed of *P. crassifolia*, are situated on shaded and semi-shaded slopes. *P. crassifolia* is a shallow-rooted species, with the majority of its roots located within the top 30 cm of the soil surface [30]; this layer constitutes 40 to 60% of the total active profile moisture storage in temperate climate regions [32]. Grasslands are predominantly found on sunny and semi-sunny slopes, while shrublands are located in the transitional zones between forests and grasslands. Shrub species include *Sabina vulgaris* Ant., *Salix oritrepba* Schneid., *Berberis kansuensis* Schneid., *Potentilla glabra* Lodd., *Potentilla*

*fruticosa* L., and herbaceous species like *Elymus cylindricus* (Franch.) Honda, *Achnatherum splendens* (Trin.), are primarily located on sunny, south-facing, and semi-shaded east- or west-facing slopes. Since the 1970s, extensive deforestation at lower elevations has occurred to meet the growing demands for timber and agricultural land in the region. Dahuang Mountain has been incorporated into the Natural Forest Protection Project since 2001 to combat deforestation-induced flooding and most of the grasslands on east- and west-facing slopes have been progressively transformed into *P. crassifolia* plantation forests. Between 2000 and 2010, the forested area in the study region expanded by 633.8 hectares [33].

*2.2. Experimental Design and Sampling*

2.2.1. Experimental Plots

A total of ten hillslope plots, ranging in size from 800 to 4000 m$^2$ and at elevations between 2595 and 2683 m, were established from 26 July to 1 August 2014, along two transects (horizontal or vertical axis) within an 85-hectare area of the Dahuang Mountain Forest Reserve (Figure 2). Two plots were situated on the north-facing slope, while the remaining plots were on northwest and northeast-facing slopes, all characterized by land use heterogeneity.

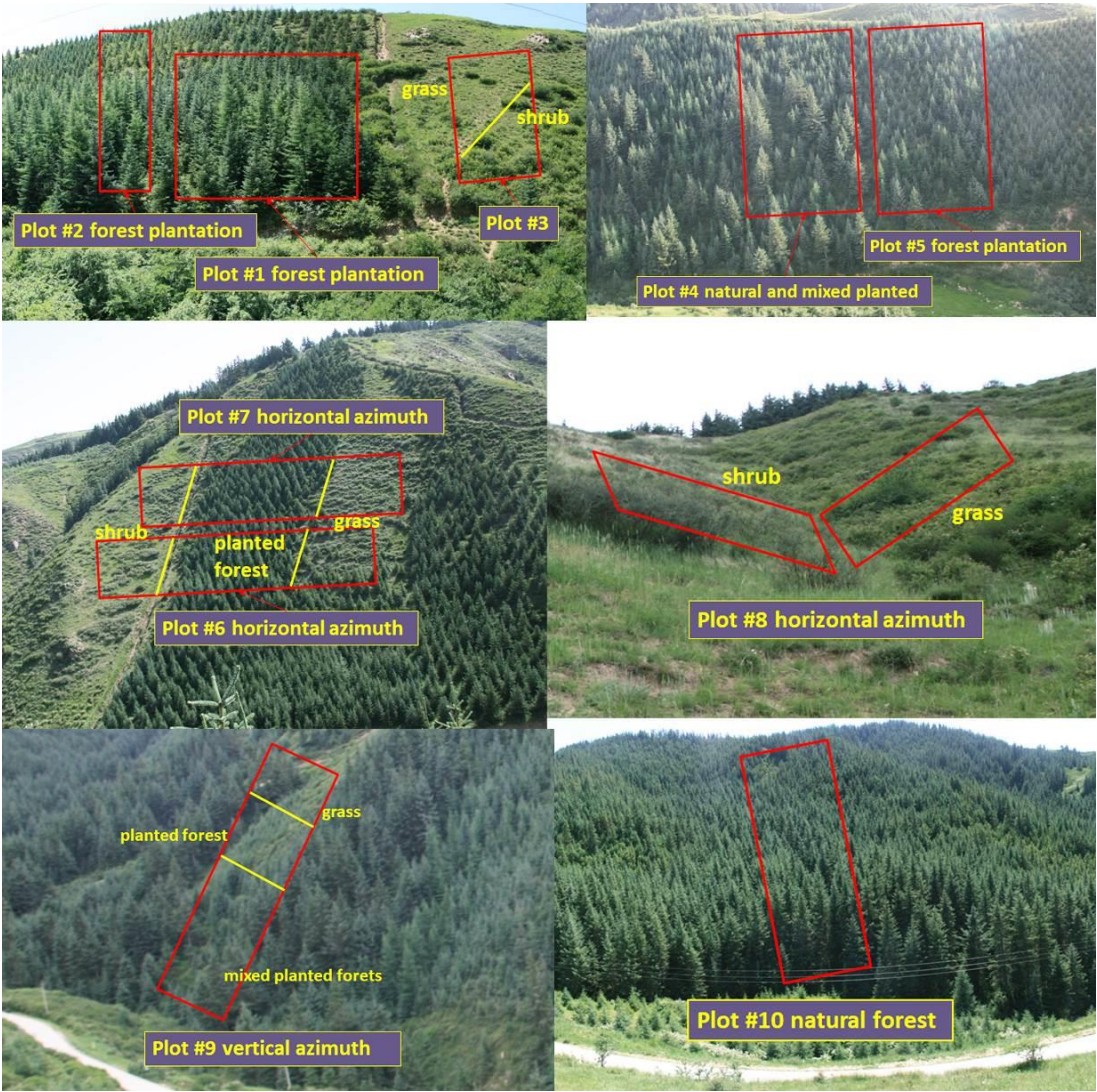

**Figure 2.** Pictures of study plots and their layout of land cover along the slope-sides. Yellow lines in the figure demarcate the different vegetation types within a plot.

Plots situated along the horizontal axis (plots #6, #7, and #8) were utilized to ascertain the relationship between land cover and soil moisture, where differences in soil moisture attributable to slope position could be excluded. Plots along the vertical axis (plots #1, #2, #3, #4, #5, #9, and #10) were employed to analyze the spatial heterogeneity relationship between soil moisture and vegetation types, taking into account slope position factors. Five spatial sampling intervals (#5, #6, #7, #8, and #10 m) were implemented for the investigation; all sampling maintained a consistent interval distance at each transect to adhere to the theoretical model of semivariance analysis. Four replicate measurements of soil moisture were recorded at each sampling point and these were averaged for the point after the exclusion of outlier values (Table 1). Concurrently with the soil moisture survey, stand structure, soil texture, and land cover species were also assessed.

### 2.2.2. Soil Water Content (SWC)

Soil water content at a volumetric level was quantified utilizing Time Domain Reflectometry (TDR) (IMKO, Ettlingen, Germany) at a depth of 16 cm. The calibration of Time Domain Reflectometry (TDR) probes was conducted in comparison with soil moisture content determined gravimetrically, which served as the benchmark reference values. The calibration error between the readings obtained from the instrument and the reference values was found to be less than 2%. A comprehensive sampling encompassing 480 points was executed over a period of six consecutive days, spanning from the 27th of July to the 1st of August. The sampling protocols mandated a minimum of seven consecutive days of sunshine and the absence of precipitation during the measurement period. This requirement was established to minimize the potential impact of precipitation on soil moisture content across various vegetation plots. Consequently, this approach aimed to enhance the accuracy of the subsequent analysis that explores the relationship between soil moisture content and environmental variables, including vegetation type.

### 2.2.3. Statistical Analysis

The statistical software SPSS 19.0 (SPSS Inc., Chicago, IL, USA) was employed to discern disparities in soil moisture across various land cover types. Prior to semivariance analysis, all datasets were subjected to normality assessment and consistency checks across land cover types using the One-Sample Kolmogorov–Smirnov Test within SPSS. In areas defined by a homogeneous vegetative composition, a unique suite of assessment metrics was established. In contrast, for areas encompassing diverse vegetative compositions, distinct assessment metrics were determined for each type of vegetation, as illustrated in Figure 2. Comparative analysis of these parameters was conducted using a one-way ANOVA to more precisely quantify the surface soil water content associated with each vegetation type. Zonal statistics analysis was employed on composite plots, specifically plots #3, #6, #7, and #9, following the demarcations illustrated in Figure 2. The descriptive statistical measures for soil moisture across each type of land cover encompassed the mean, standard deviation, coefficient of variation, minimum, maximum, skewness, and kurtosis. Regression analyses and graphical representations were generated using OriginPro 9.0 (OriginLab, Northampton, MA, USA), with data preprocessing facilitated by Microsoft Excel 2016.

To elucidate the inherent interactions between soil water content (SWC) and all pertinent environmental factors, including plot number (ID), vegetation type (VEG), forest age (AGE), slope position (SP), slope gradient (SLP), slope aspect (SLPA), vegetation height (HIG), vegetation density (DEN), survey plot area (ARE), and sample size (NUM), encompassing a total of nine variables, the Generalized Additive Mixed Model (GAMM) was utilized for modeling and analysis of the survey data. A Generalized Additive Mixed Model (GAMM) extends the generalized linear mixed model (GLMM) framework by incorporating smooth functions of certain predictor variables. These smooth functions, often referred to simply as "smooths", allow for the modeling of non-linear relationships between the predictors and the response variable.

**Table 1.** Basic description of the 10 study plots.

| Plot ID | Plot Size (m) | Horizontal Sampling Interval (m) | Vertical Sampling Interval (m) | Number of Survey Samples | Elevation (m) | Slope Gradient | Slope Aspect | Slope Aspect | Age of Stand | Height of Tree (m) | Density of Stand (ha$^{-1}$) | The Primary Vegetation Type of the Plot |
|---|---|---|---|---|---|---|---|---|---|---|---|---|
| #1 | 80 × 50 | 10 | 5 | 99 | 2595 | 31 | 290 | Half-shady slope | 45 | 6.3 | 3192 | Planted forest plot, single slope surface |
| #2 | 21 × 72 | 7 | 8 | 40 | 2595 | 32 | 74 | Half-shady slope | 35 | 4.2 | 2833 | Planted forest plot, single slope surface |
| #3 | 20 × 40 | 5 | 8 | 30 | 2595 | 32 | 74 | Half-shady slope | 19 | 3.6 | 2725 | Grassland plot, single slope surface |
| #4 | 20 × 40 | 5 | 8 | 30 | 2600 | 47 | 43 | Shady slope | 44 | 6.7 | 2967 | Planted-natural mixed forest plot, single slope surface |
| #5 | 20 × 40 | 5 | 8 | 30 | 2595 | 36 | 74 | Half-shady slope | 27 | 4.1 | 2795 | Planted forest plot, single slope surface |
| #6 | 56 × 32 | 7 | 8 | 45 | 2599 | 39 | 290 | Half-shady slope | 33 | 4.7 | 2917 | Shrub, planted forest and grassland plot, horizontal transect |
| #7 | 56 × 32 | 7 | 8 | 45 | 2626 | 42 | 290 | Half-shady slope | 23 | 3.8 | 2750 | Shrub, planted forest and grassland plot, horizontal transect |
| #8 | 25 × 24 | 5 | 6 | 30 | 2683 | 35 | 76 | Half-shady slope | 17 | 16.5 | 2575 | Shrub and grassland plot, horizontal Shallow V transect |
| #9 | 30 × 48 | 6 | 6 | 54 | 2683 | 26 | 56 | Half-shady slope | 39 | 4.5 | 2634 | Grassland, planted and natural forest plot, single slope |
| #10 | 36 × 60 | 6 | 6 | 77 | 2660 | 39 | 9 | Shady slope | 87 | 26.8 | 921 | Natural forest plot, single slope |

Prior to modeling, we employed the variance inflation factor (VIF, package 'usdm' 2.1-7) to conduct a collinearity analysis on the driving factors, subsequently excluding the highly correlated factor tree height (HIG). The collinearity analysis function utilized was 'vifstep', with the correlation threshold parameter (th) set at 10. In order to mitigate potential analytical inaccuracies arising from the utilization of variables measured on different scales within the modeling framework, a standardization procedure was employed for all variables under consideration. This process involved the conversion of observed values into a uniform numerical scale bounded by 0 and 1.

Within the entirety of the variables, VEG, SP, and SLPA were treated as categorical variables; in the variable VEG, the values 1, 2, 3, 4, and 5 correspond, respectively, to grass, shrub, planted forest, mixed forest, and natural forest. The variable SP denotes slope position, with the values 1, 2, and 3 representing upper slope, middle slope, and lower slope, respectively. For the variable SLPA, the values 1 and 2 distinguish between sun-facing slopes and shade-facing slopes. The ID was integrated as a stochastic variable within the Generalized Additive Mixed Model, thereby formulating a model that elucidates the association between a range of ecological determinants (comprising VEG, AGE, SP, SLP, SLPA, DEN, ARE, and NUM) and SWC. The GAMM (package 'gamm4' 0.2-6) was implemented using R version 4.3.3 for Windows, with RStudio version 2023.12.1-402 serving as the computational platform.

### 2.2.4. Geostatistical Analysis

Geostatistical analysis enables the quantification of the spatial attributes of soil moisture across individual plots and allows for the discernment of potential spatial configurations by employing interpolation methods. These methods are based on the parameters obtained from the outcomes of the analysis. These techniques involve the computation of the semivariance $r(h)$ for all feasible pairs of proximate sampling points, based on their separation distance and the degree of autocorrelation among the points [12].

The geostatistical analysis comprises two principal components: (1) the assessment of the autocorrelation degree among the measured points and (2) the interpolation for analogous points not directly sampled, predicated on the encountered degree of autocorrelation. The semivariance function $r(h)$ is calculated using the following formula:

$$r(h) = \frac{1}{2N(h)} \sum_{i=1}^{N(h)} \left[ z(x_i) - z(x_{i+h}) \right]^2 \qquad (1)$$

Herein, $r(h)$ represents the semivariance for a given interval distance class $h$, $h$ denotes the sampling interval within each plot, $z(x_i)$ is the measured value at point $x_i$, $z(x_i + h)$ is the value at point $x_i + h$, and $N(h)$ is the aggregate number of sample pairs within the distance interval $h$. The configuration of the resultant plot of $r(h)$ elucidates the degree of autocorrelation present.

Autocorrelation analysis and kriging interpolation were executed using GS+ 7.0 (Geo-Statistics for the Environmental Sciences, GS+, Gamma Design Software, Plainwell, MI, USA). Data acquisition was conducted on a bi-dimensional lattice situated on an incline of a mountain. To delineate and ascertain the primary axis of anisotropy, which corresponds with the orientation exhibiting the minimal semivariance for each section, an azimuth function was utilized. Four isotropic models, linear, spherical, exponential, and Gaussian are available for characterizing the semivariance versus separation distance relationship [34]. Three parameters are essential for modeled variograms: (a) the separation distance over which spatial dependence is discernible (range, $A$), (b) the level of inherent randomness (nugget variance, $C0$), and (c) the total variation present (sill, $C + C0$). The ratio $C/(C + C0)$ quantifies the spatial dependence of soil moisture, with values greater than 0.75 indicating strong spatial dependence, values between 0.25 and 0.75 denoting moderate spatial dependence, and lower values suggesting a weaker autocorrelation [35,36]. The initial lag distance for semivariogram analysis for each plot was set to 0.5 m of the sampling interval.

The nature of spatial variability in the data is depicted by the four variogram models [37,38]. To evaluate the performance of the statistical model, the Residual Sums of Squares (*RSS*) was employed as a measure of the model's fit to the variogram data. *RSS* is a sensitive and robust index and it also serves as the parameter for the variogram model obtained by determining a combination of parameter values. Thus, semivariogram model-fitting was performed based on the lowest *RSS*. Additionally, regression and the coefficient of determination ($r^2$) were utilized to indicate the model's fit to the variogram data. Kriging interpolation maps for each mean soil moisture interval were computed to describe the spatial variability patterns of soil moisture.

## 3. Results

### 3.1. Soil Moisture Variability within Land Cover Types

Data pertaining to soil moisture, collected from 10 sampling plots across diverse land cover types, are presented in Figure 3 and Table 2. The data for each plot adhered to a normal distribution, as evidenced by the absolute values of skewness and kurtosis being less than 1. Notable variations in surface soil moisture, ranging from 9.01 to 16.55%, were observed among the different land cover types. For instance, the soil moisture in shrubland, recorded at 15.81%, was significantly higher than that in planted forest and grassland, which were 12.28% and 13.30%, respectively (refer to plot #6 in Figure 3). Conversely, the soil moisture levels observed in grassland areas did not demonstrate a statistically significant variance in comparison to those recorded in both planted and mixed forest ecosystems, which exhibited a moisture content of 10.53% (as indicated in plot #9 of Figure 3). The disparity in soil moisture between natural forests, at 16.5%, and planted forests, at 11.82%, was statistically significant, as indicated by the mean values of plot #10 and plot #2.

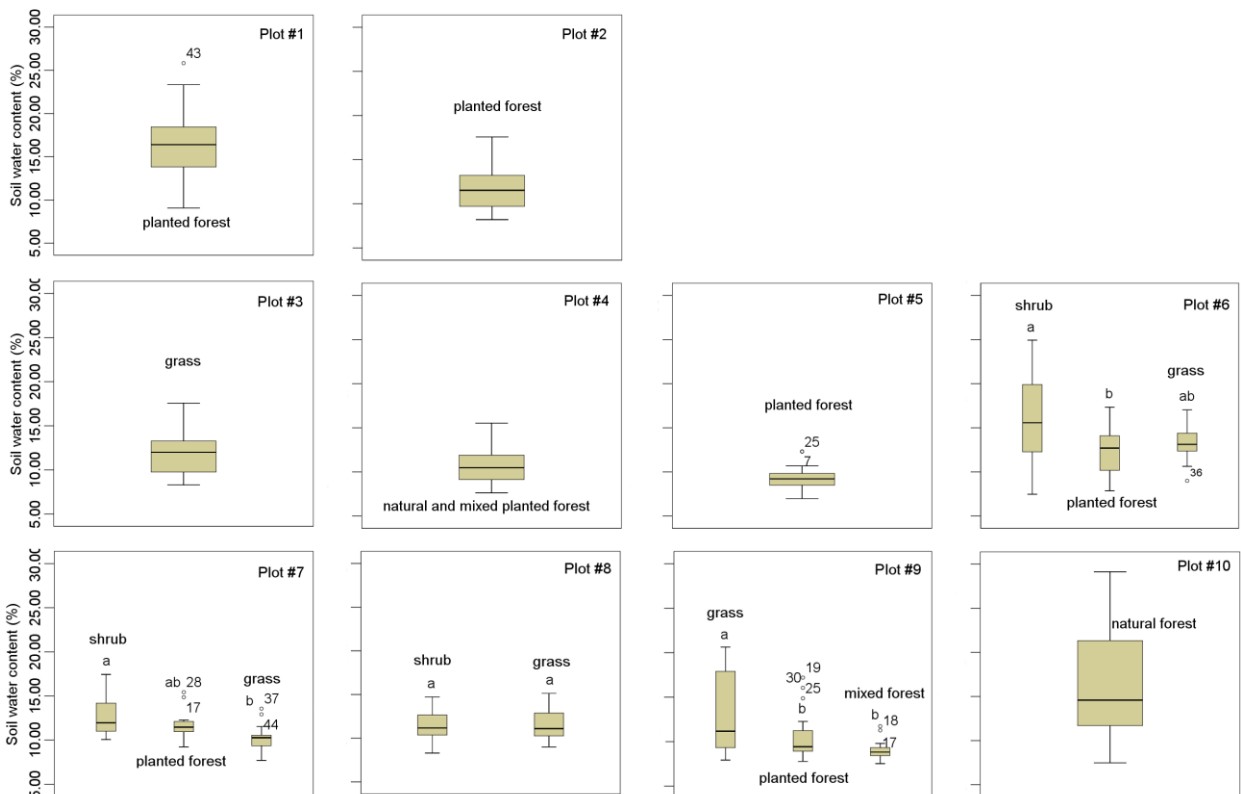

**Figure 3.** Soil water content across different plots, employing boxplots that depict the median value, lower and upper quartiles, minimum and maximum values, outliers, and the level of statistical significance based on vegetation type. The annotations 'a', 'ab', and 'b' denote the significance levels among the land covers within a single plot, while 'o' represents the potential outlier value for the land cover type, accompanied by the label of the value.

**Table 2.** SWC (%) for each sample plot, categorized by land cover type, serves as a quantification and supplement to Figure 3.

| Plot ID | Vegetation Type | Number of Sampling Points (*n*) | Mean | Std. Deviation | Minimum | Maximum | Skewness | Kurtosis | Variance |
|---|---|---|---|---|---|---|---|---|---|
| #1 | Planted forest | 99 | 16.49 | 3.59 | 9.07 | 25.83 | 0.29 | −0.56 | 12.88 |
| #2 | Planted forest | 40 | 11.82 | 2.55 | 8.20 | 17.57 | 0.75 | −0.08 | 6.49 |
| #3 | Grass, Shrub | 30 | 12.11 | 2.62 | 8.30 | 17.57 | 0.71 | −0.16 | 6.84 |
| #4 | Natural and mixed planted forest | 30 | 10.91 | 2.15 | 7.63 | 15.53 | 0.68 | −0.28 | 4.64 |
| #5 | Planted forest | 30 | 9.26 | 1.27 | 6.98 | 12.33 | 0.53 | 0.82 | 1.61 |
| #6 | Shrub | 15 | 15.81 | 4.99 | 7.47 | 24.93 | 0.22 | −0.80 | 4.99 |
| #6 | Planted forest | 15 | 12.28 | 2.74 | 7.87 | 17.34 | 0.21 | −0.75 | 7.50 |
| #6 | Grass | 15 | 13.30 | 2.08 | 9.00 | 17.03 | −0.02 | 0.62 | 4.33 |
| #7 | Shrub | 15 | 12.68 | 2.28 | 10.07 | 17.44 | 0.73 | −0.49 | 5.20 |
| #7 | Planted forest | 15 | 11.71 | 1.66 | 9.23 | 15.43 | 0.99 | 1.31 | 2.75 |
| #7 | Grass | 15 | 10.18 | 1.61 | 7.70 | 13.57 | 0.58 | 0.58 | 2.60 |
| #8 | Shrub | 15 | 11.54 | 1.81 | 9.00 | 15.13 | 0.38 | −0.68 | 3.28 |
| #8 | Grass | 15 | 11.39 | 1.84 | 8.30 | 14.72 | 0.30 | −0.52 | 3.39 |
| #9 | Grass | 18 | 13.35 | 4.56 | 7.93 | 20.63 | 0.38 | −1.57 | 20.84 |
| #9 | Planted forest | 18 | 10.53 | 2.77 | 7.77 | 17.20 | 1.50 | 1.25 | 7.68 |
| #9 | Planted-natural mixed forest | 18 | 9.01 | 1.08 | 7.53 | 11.73 | 1.31 | 1.87 | 1.17 |
| #10 | Natural forest | 77 | 16.55 | 6.16 | 7.48 | 29.13 | 0.56 | −0.77 | 37.90 |

To further elucidate the disparities in soil moisture among the land cover types, the data were aggregated and analyzed using SPSS, categorized by land cover type (as shown in Figure 4). The results indicated a descending trend in soil moisture from natural forest, through shrubland, grassland, and planted forest, to mixed forest. The soil moisture levels in natural forests and mixed forests were significantly different from those in other vegetation types, with a *p*-value of less than 0.05. The soil moisture in mixed planted forests was approximately 25% lower than that in shrub, grass areas, and planted forests, whereas it was higher in natural forests compared to other vegetation types.

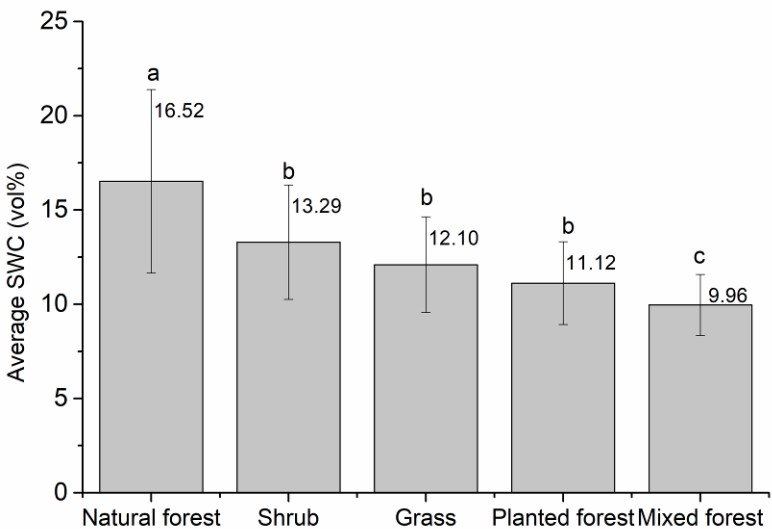

**Figure 4.** Soil water content (SWC) for each vegetation type across the ten study plots, showcasing average values and standard deviations. Different lowercase letters indicate significant differences between vegetation types at a *p*-value of 0.05 or less.

Moreover, the standard deviations in natural forest (plot #10) and planted forest (plot #1) were 6.16% and 3.5%, respectively, while in shrubland and grassland (plot #8 and #3), they were 1.18% and 2.62%, respectively, as reported in Table 2. A larger standard deviation suggests a greater heterogeneity in soil water content at the forest floor compared to that in shrublands and grasslands.

### 3.2. The Semivariance Characteristics of Soil Moisture in the Sampling Plots

Variogram analysis and Kriging interpolation were utilized to characterize and generate spatial patterns of soil moisture for each plot, as detailed in Table 3. *RSS*, for most plots, was less than 1 (with an average value of 0.23 for plots #2 and #7), indicating a pronounced spatial auto-correlation among the data distribution within their effective ranges. This implies that the models utilized for semivariance analysis were suitable and that the parameter combinations for each fitting function model can be employed to elucidate the spatial patterns observed in the experimental plots. However, the *RSS* for plot #1 was 12.9%, implying that the model used for this plot did not fit as well as theoretically required and thus the interpolation map should be interpreted with caution in subsequent analyses.

**Table 3.** Variogram models and parameters for soil moisture (vol%) in each sampling plot, including *RSS* and the coefficient of determination ($r^2$).

| Plot ID | Variogram Model | Nugget Variance $C_0$ (m) | Sill $C_0 + C$ (m) | Spatial Correlation $C/(C + C_0)$ | Range $A$ (m) | *RSS* | $r^2$ | Sampling Interval (m) | Orientation |
|---------|-----------------|---------------------------|--------------------|-----------------------------------|---------------|-------|-------|-----------------------|-------------|
| #1 | Linear | 12.85 | 13.235 | 0.03 | 51.47 | 12.90 | 0.01 | 10 | Vertical azimuth |
| #2 | Spherical | 2.40 | 9.280 | 0.74 | 47.28 | 0.01 | 0.99 | 7 | Vertical azimuth |
| #3 | Gaussian | 0.11 | 6.848 | 0.98 | 4.31 | 0.00 | 0.99 | 5 | Vertical azimuth |
| #4 | Exponential | 2.88 | 6.252 | 0.54 | 26.2 | 0.66 | 0.75 | 5 | Vertical azimuth |
| #5 | Spherical | 0.00 | 1.541 | 1.00 | 7.83 | 0.07 | 0.00 | 5 | Vertical azimuth |
| #6 | Spherical | 0.39 | 14.100 | 0.97 | 12.74 | 0.36 | 0.86 | 7 | Horizontal azimuth angle |
| #7 | Exponential | 0.01 | 3.924 | 1.00 | 6.69 | 0.31 | 0.61 | 7 | Horizontal azimuth angle |
| #8 | Linear | 2.99 | 2.990 | 0.00 | 30.46 | 1.50 | 0.00 | 5 | Horizontal azimuth angle |
| #9 | Exponential | 1.74 | 16.260 | 0.89 | 20.66 | 0.20 | 0.99 | 6 | Vertical azimuth |
| #10 | Spherical | 0.01 | 27.090 | 1.00 | 13.59 | 2.01 | 0.98 | 6 | Vertical azimuth |

In the spatial modeling of different sample sites, a variety of fitting functions were employed. For example, plot #3 used a Gaussian model, while plots #2, #5, #6, and #10 utilized a spherical model. In contrast, plots #4, #7, and #9 employed an exponential model, indicating that the fitting model was robust and the interpolation results were based on a function capable of representing the actual spatial pattern of soil moisture. The optimal function was linear for plots #1 and #8, which suggested that the model was not as ideal as indicated by the *RSS*; this may be a limitation due to the dataset size. A spatial correlation of less than 0.5 between soil moisture in plots #1 and #8 indicated an absence of spatial correlation, which was consistent with the *RSS* findings. Other sampling plots exhibited strong spatial auto-correlation, suggesting that the parameters based on the models are suitable for spatial interpolation of soil moisture.

As depicted in Figure 5, the differences in soil moisture content among all the plots were not significant, particularly for plots #5, #6, and #7. The spatial heterogeneity was more pronounced in plot #1, an artificial forest plot, and plot #10, a pure natural forest plot, with both plots exhibiting higher soil moisture content at the downslope positions. Although plots #3, #4, and #8 exhibited a certain degree of spatial heterogeneity, the spatial variability was significantly attenuated due to the constraints of a uniform legend.

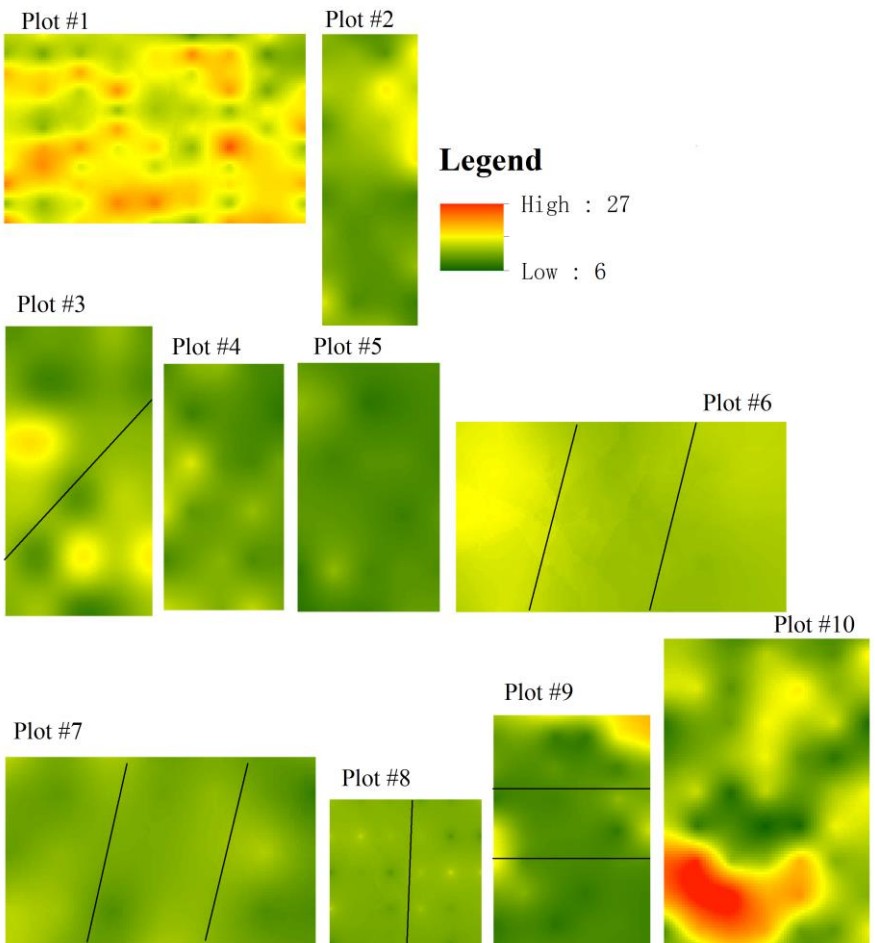

**Figure 5.** Kriging interpolation maps of soil moisture for the ten study plots in Dahuang Mountain. The spatial pattern in each plot corresponded with the layout of land cover types as depicted in Figure 2, with different colors indicating the SWC (vol%).

Analysis revealed that within the Generalized Additive Mixed Models (GAMM), the family was identified as Gaussian with an R-squared value of 0.32. The log-likelihood value using Restricted Maximum Likelihood (REML) for the linear mixed-effects model (lmer) was $-441.93$ and the scale estimate stood at 0.02. The sample size for this study was 480. The final model employed was the Linear Mixed-Effects Model (lmer).

The results from Table 4 indicate that factors such as vegetation type (VEG), slope position (SP3), aspect of shady slope (SLPA2), and plot size (ARE) sample quantity (NUM) have a significant impact on the moisture content of the surface soil ($p < 0.01$). Among these, the mixed forest (VEG4), lower slope position (SP3), aspect of shady slopes (SLPA2), and plot size (ARE) exhibit particularly significant effects ($p < 0.001$).

**Table 4.** Parametric coefficients of the SWC and environmental factors based on GAMM.

| Variable | Estimate | Std. Error | t Value | Pr(>\|t\|) | Independent Variable Name |
|---|---|---|---|---|---|
| Intercept | 0.31 | 0.10 | 3.15 | $1.74 \times 10^{-3}$ ** | Intercept |
| VEG2 | $-0.04$ | 0.10 | $-0.41$ | 0.683 | Shrub |
| VEG3 | $-0.31$ | 0.12 | $-2.66$ | 0.008 ** | Planted forest |
| VEG4 | $-0.55$ | 0.16 | $-3.41$ | 0.001 *** | Mixed forest |
| VEG5 | $-0.73$ | 0.27 | $-2.69$ | 0.008 ** | Natural forest |
| AGE | 0.19 | 0.29 | 0.65 | 0.515 | Forest age |
| SP2 | 0.04 | 0.02 | 2.34 | 0.020 * | Middle slope position |

**Table 4.** *Cont.*

| Variable | Estimate | Std. Error | t Value | Pr(>|t|) | Independent Variable Name |
|---|---|---|---|---|---|
| SP3 | 0.08 | 0.02 | 4.04 | $6.20 \times 10^{-5}$ *** | Lower slope position |
| SLP | −0.18 | 0.07 | −2.52 | 0.012 * | Slope gradient |
| SLPA2 | 0.38 | 0.09 | 4.07 | $5.51 \times 10^{-5}$ *** | Shade slope |
| DEN | −0.08 | 0.10 | −0.81 | 0.420 | Vegetation density |
| ARE | 0.34 | 0.07 | 5.13 | $4.24 \times 10^{-7}$ *** | Area of plot |
| NUM | 0.18 | 0.07 | 2.73 | 0.007 ** | Number of the tree |

Note: Significant codes are as follows: 0–0.001: ***, 0.001–0.01: **, and 0.01–0.05: *.

## 4. Discussion

### 4.1. Variability of Soil Moisture across Different Land Cover Types

Precipitation levels typically dictate the predominant land cover type and its spatial distribution within an ecosystem [19,39]. Nonetheless, anthropogenic activities alter the dynamics of this relationship, particularly when large-scale environmental programs, such as China's "Grain for Green," are implemented [27,40]. The introduction of new land cover types modifies the hydrological cycle through abiotic and biotic processes, namely evaporation and transpiration. This complex interplay between land cover and environmental factors engenders a gradual emergence of trade-offs, which can have lasting and irreversible effects on the livelihoods of local populations [39].

The mean soil moisture in the natural forest examined in this study (16.49 ± 3.59%) was approximately 72% lower than that reported by Zhu [41] (23 ± 4%) in an unthinned *P. crassifolia* stand during the 2012 growing season at Guantai. The observed variation in soil moisture content could be ascribed to the differing durations of antecedent drought conditions preceding the data collection. It is hypothesized that reduced temporal gaps between rainfall occurrences before the assessment may result in an inflated estimation of the mean soil moisture levels. Additionally, the observation that SWC in mixed forests is lower than in plantations (Figure 4) suggests that after a portion of a plantation forest that has been cultivated for approximately a decade dies, reforestation is conducted through supplementary planting. Consequently, the lower SWC in mixed forests may indicate an objective phenomenon of reduced water availability that could have occurred during the plantation phase [42]. This finding contrasts with those of several prior studies [26,41] and may reflect the repetitive degradation of the planted area over the past two decades [8,19,43–45]. Initially, there may be a marginal decrement in pedospheric hydric content attributable to the augmented hydric requisites of arboreal proliferation and the desiccation of exposed pedologic surfaces. Subsequently, a novel homeostasis between vegetative hydric consumption and precipitation inputs is established, encompassing arboreal demise as a consequence of diminished pedospheric moisture. In due course, a relatively stable hydrologic cycle is constituted [10,19,46].

In Table 4, the investigation clarified the paramount environmental determinants influencing the surface soil moisture levels in arid zones. These encompass the establishment of synthetic woodlands, the inferior elevation on inclines, the orientation of slopes receiving less sunlight and the observational scale denoted by the dimensions of the examination territory. It is evident that the succession of artificial forests indeed induces significant changes in soil moisture content; however, the influence of micro-topography, such as slope position and aspect, is also of considerable importance. It is particularly noteworthy that the scale of investigation can have a substantial impact on the magnitude of soil moisture content, necessitating close attention in similar experimental endeavors. Our research indicates that forested areas in arid zones have the capacity to accrue soil moisture over an extended period following the establishment of plant life, contingent upon the vegetation reaching full maturity. This underscores the ecological efficacy of afforestation efforts, which exhibit long-term hysteresis phenomena [43,47].

Topography is another factor that can contribute to variability in soil moisture [22,46]. The block kriging method used to create the soil moisture kriging map in this study can only reveal the spatial variability of soil moisture within a single plot and is not suitable for comparing soil moisture across different plots (Figure 2 and Table 2). However, a significant trend of increasing soil moisture was observed at the bottom of slopes within a single vegetation type, as shown in Table 4. As observed in Figure 5, only a few sample sites exhibit relatively pronounced topographic heterogeneity, such as plots #4 and #10. The majority of the other plots did not exhibit significant differences in soil moisture content across slope positions (plots #2 and#9). This phenomenon can be attributed to a runoff depth decrease of 14.3%, which suggests that forests might induce a diminution in the net water yield, given an average annual precipitation of 374.1 mm in the Pailugou catchment that is proximate to the area under investigation [48]. This implies a lack of subsurface lateral flow within this particular region. These results diverge from those of other studies [7,49], which found that soil moisture is influenced by subsurface lateral water flow from the upper slope to the toe of the slope under conditions of adequate water supply. The absence of such moisture flow precludes the formation of self-organized vegetation patterns, as reported in the loess hilly region of China by Sun [19]. Moreover, no spatial correspondence was found between soil moisture and vegetation types, particularly in plots #6 and #7 and the spatial distribution of soil moisture was consistent with slope aspect (Table 4). Terrain-related processes may be the primary environmental factor influencing the spatial pattern of vegetation during wet periods [32,50] but this was not observed in the current study.

### 4.2. Spatial Pattern and Heterogeneity of Soil Moisture across Succession Stages of a Planted Forest

The spatial auto-correlation indicator $C/(C + C_0)$ exceeded 0.75 for most plots, indicating a continuous spatial pattern of soil moisture among the sampling plots (Table 3). The study revealed spatial heterogeneity in the auto-correlation length (A) of soil moisture, with forested and mixed forest areas demonstrating a wider variability in soil moisture relative to grasslands and shrub-dominated ecosystems. Notably, the auto-correlation length in plot #3, which was 4.3 m, is documented in Table 3. Additionally, *A* for planted or mixed forests was larger than that for natural forest areas, suggesting that soil moisture is less stable in planted land cover during the initial decades of growth [29,43,51]. The residual sum of squares (*RSS*) for the fitting models approached zero, signifying a significant spatial dependence as determined through geostatistical analysis. This phenomenon could be linked to simultaneous solar irradiation and evapotranspiration influencing soil moisture across various land cover types [18,52]. The observed minimal nugget variance value of 0.01 ($C0$) in conjunction with the maximal sill value of 27.09 ($C + C0$) implies that natural forests, as denoted by plot #10 in Table 3, exhibit the highest potential to diminish spatial heterogeneity in surface soil moisture. Furthermore, these forests demonstrate a notable ability to constrain variations in soil moisture content, thereby indicating a substantial capacity for soil water retention relative to other vegetation types. Conversely, the minimal sill value of $C + C0$, averaging 7.6 across plots #1, #2, #5, and #6, suggests that the plantation's surface soil moisture exhibits limited variability, denoting a comparatively inferior water retention capability.

Interpolation techniques delineate the spatial distribution of soil moisture in a manner that is more readily comprehensible. The parcels of land characterized by diverse plant species, denoted as #4 and #9 in Figure 5, demonstrated significant autocorrelation, with a uniform spatial distribution of soil moisture. This pattern suggests that in these areas, soil moisture levels are primarily influenced by the frequency and intensity of precipitation, as corroborated by the literature cited in references [46,52,53]. Additionally, the plantation of artificial forests exerts a significant impact on the soil moisture content in arid regions, as suggested by reference [37]. In the intervals between precipitation occurrences, the location on a slope and the direction it faces predominantly dictate the spatial distribution of surface soil moisture, as evidenced in Table 4 [54]. It is essential to recognize that owing

to the marked spatial variability in soil moisture content within arid zones, the scale of the research plots and the number of samples collected are intimately linked to the resultant values of soil moisture content.

## 5. Conclusions

Comprehending the spatial dynamics of soil water content throughout the succession of artificial forests and identifying the principal factors that influence its variability is critical for enhancing our understanding of hydrological and biogeochemical cycles, particularly in basins without gauging stations. The current study has demonstrated a significant reduction in soil moisture content, progressing from natural forests to shrublands, then to grasslands, and finally to planted and mixed forests. This gradient reflects the pattern of surface soil moisture modification throughout the succession of plant communities in arid regions. The outcomes of the Generalized Additive Mixed Model analysis suggest that in semi-arid and arid regions, the duration of afforestation, lower slope position, shady slope, and the scale of investigation all exhibit a high degree of correlation with soil moisture content. The study's findings indicate that in these regions, planted and mixed forests aged between 30 and 50 years exhibit reduced soil moisture levels in comparison to natural forests or shrublands. Future research should aim to elucidate the interactive mechanisms between the spatial distribution of soil moisture and vegetation patterns on slopes.

The findings of this research can guide the development of strategies aimed at improving the enduring success of re-vegetation projects within the region. For example, prioritizing the establishment of herbaceous or shrubby vegetation over arboreal species could mitigate the high rates of tree mortality attributed to inadequate soil water content (SWC). This approach may also shed light on the factors contributing to the extensive mortality observed in afforested areas. In theory, the spatial distribution and volume of SWC in a plantation could provide critical reference points for formulating more empirically based irrigation and cultivation tactics for the stewardship of plantations in these environments.

**Author Contributions:** J.Y.: Conceptualization, Software, Visualization, Writing—review & editing. L.G.: Conceptualization, data processing and graphical production. Y.L.: Review and editing. P.L. and J.D.: Experimental layout, data acquisition, and model construction. All authors have read and agreed to the published version of the manuscript.

**Funding:** This study was funded by the National Natural Sciences Foundation of China (41901050, 41901044 and 32271667).

**Data Availability Statement:** Data will be made available on request.

**Acknowledgments:** The authors extend their gratitude to the two peer reviewers for their comprehensive recommendations, which have significantly enhanced the objectivity and scientific rigor of the discussion section within the manuscript.

**Conflicts of Interest:** The authors declare no conflict of interest.

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
