# Peer review of "Reforestation Will Lead to a Long-Term Downward Trend in the Water Content of the Surface Soil in a Semi-Arid Region"

_forests, doi:10.3390/f15050789_

Round 1

Reviewer 1 Report

Comments and Suggestions for Authors

General comments

While this manuscript contains some interesting and useful information, it needs to revised to make a number of things clearer.

This study examines spatial variation over a limited time period.  That is of use, but it needs to be clear what aspect of variation is being examined (spatial and not temporal). It also needs to be made clear why the particular period was selected and what the results mean in the context of changes throughout the year.

The connection between soil moisture in the period sampled, succession, and future changes in the water balance of these systems is not clear.  They all seem to be mixed together and that is not only confusing, but it reduces the impact of the findings.  Also succession seems to combine the effects of land cover conversion and what many would consider succession.  Those ideas also need to be separated and clarified.

The concerns about soil moisture are also mixed together leading to confusion.  Is the concern about the soil moisture related to the success of reforestation or afforestation efforts?  Or is it about water yield? Or whether was use by land cover types will lead to their own demise in the future?  These are all separate concerns, but they all seem to be mixed together.

The context for this study is very narrow geographically.  Concerns about the effects of land cover conversion are likely global.  Concerns about the effects of changing climate and reduction in water availability are also global in nature.  The authors need to improve the global context and importance of this study.

The spatial geostatistics analysis does add some degree of novelty to the manuscript.  However, it is difficult to match the patterns in Figure 5 to the locations of the various cover types.  It would be very helpful to somehow indicate the location of the cover types on that figure. This can be done to some degree by going to Figure 2, however the perspective differs and it is not particularly convenient.

There are a number of wording problems described below that need to be addressed. Also the formatting of the tables makes them unreadable. 

Specific comments (line)

2 I don’t think that the title reflects the conclusions correctly. It seems to imply that there are no downward fluctuations in non-forested cover types.  However, given seasonal variations all cover types have downward fluctuations as they have upward fluctuations.  What seems to have been found is that reforestation leads to a long-term downward trend.

5 I don’t believe that succession is the correct word here.  Succession refers to the long-term trend in vegetation composition and structure.  As the rest of the abstract does not refer to that I can only assume that the correct word is success.

7 this makes it seem as if there are two entities a native tree and a dominative species.  I am assuming they are one and the same?

9 The way this is written it seems to suggest the concern about soil moisture is whether it is sufficient to plant trees.  However, the thrust of the uncertainty seems to be whether reforestation has had impacts on soil moisture.  I suggest the authors revise to make the uncertainty that is the focus of the analysis clearer.

19 This conclusion is based on an unspecified management objective.  Please specify what the objective is for the reader.

39 If the authors already know that planting of this species will lead to these changes, then why conduct the analysis described in the manuscript?

54-56 This section seems to compare two entities, but it is not clear what they are.

62 controlling factor?

65 I am not sure what “the main zone” means.

68 I am not sure the logic behind this statement is true.  It seems to leave a number of important factors out.

80 understanding?

92 is evolution needed in this sentence?

102 is the word perfect needed? Perfect is a very high standard to attain.

121 Would “at the lower elevations” make more sense? Or was it the lowest elevation? As written it is not clear.

125 Is there a more concise way to express this?  I take it that the average ratio of forest to grass cover increased 6.2% over the period in question.

149 the formatting of the table makes it difficult to read the headings.

166 In order is not needed in any sentence. Please remove. Also why was this time period selected other than the precipitation conditions?  Does this represent a season low point or average? Or an annual one? It is not clear why.

179 Please move results to the results section.

185 Again table formatting makes it difficult to read.

186-224 This section is written more along the lines of a textbook than a methods section.  Providing some background to reads is fine, but this has to primarily focus on the actual methods used.

242 What is the mixed forest? Do the authors mean the natural and planted mixed forest?  Please be consistent in use of terms so the reader can understand the results.

268 Again table formatting makes it hard to read.

269 This reads like a set of notes.  Please revise using complete sentences.

291 Hasn’t this been known for a longer period than since 2014? And in a wider range of locations? Wasn’t this the basis of Clements’ climax?

301-305 I found this sentence confusing because it combines several ideas that are difficult to sort out as written.

326 I am not sure that “in other words” makes sense in this context.  That implies there will be an alternative wording, not an elaboration.

364 On the other hand and but mean the same thing.  Use one or the other, but not both.

369 Understanding?  Also this seems to state that understanding improves understanding.  That reads quite oddly.

371 It should be made clear that spatial and not temporal variability was examined.

Comments on the Quality of English Language

See specific suggestions in review comments

Author Response

General comments:

  1. Comment: “While this manuscript contains some interesting and useful information, it needs to revised to make a number of things clearer.”

Response: Thank you very much for your recognition of our work. We will complete the revisions one by one according to your opinions.

  1. Comment: “This study examines spatial variation over a limited time period. That is of use, but it needs to be clear what aspect of variation is being examined (spatial and not temporal). It also needs to be made clear why the particular period was selected and what the results mean in the context of changes throughout the year”.

Response: the content of the manuscript mainly focuses on the change trend and key characteristics of soil moisture at different time stages after plantation. We have supplemented and improved the surface vegetation characteristics of different successional stages of artificial forests in the "Introduction" section at the end of the second paragraph. This provides a fundamental basis for the quantification and analysis of the soil moisture characteristics of different vegetation types in different successional stages of artificial forests in our subsequent text, which is also the basis for our exploration of the relationship between different successional stages of artificial forests and surface soil moisture. For the purpose of this study, we need to have certain prerequisites in order to conduct the investigation: that is, to find identical weather conditions that can reflect the soil moisture status corresponding to different vegetation types. In this study, prior to conducting the investigation, we happened to encounter a continuous 7-day period of sunny weather. Subsequently, we conducted research on the surface soil moisture corresponding to different vegetation types in the study area, laying the foundation for quantifying and exploring the variation patterns of soil moisture in artificial forests at different successional stages. Therefore, our research results are not related to the seasonal variation of soil moisture content corresponding to the same vegetation type.

  1. Comment: “The connection between soil moisture in the period sampled, succession, and future changes in the water balance of these systems is not clear. They all seem to be mixed together and that is not only confusing, but it reduces the impact of the findings. Also succession seems to combine the effects of land cover conversion and what many would consider succession. Those ideas also need to be separated and clarified”.

Response: We are very grateful for your suggestions. In our previous manuscript, the statement regarding this aspect was indeed not clear enough. Therefore, in this revision, we have provided a statement on the process of artificial forest succession at the end of the second paragraph of the "Introduction." In the final paragraph, we have described the artificial forest succession stages corresponding to different vegetation types surveyed in this study. In conjunction with the content in the penultimate paragraph, we have clearly explained the process and purpose of this study.

  1. Comment: “The concerns about soil moisture are also mixed together leading to confusion. Is the concern about the soil moisture related to the success of reforestation or afforestation efforts? Or is it about water yield? Or whether was use by land cover types will lead to their own demise in the future? These are all separate concerns, but they all seem to be mixed together.”

Response: I apologize for the confusion caused to you, our statements regarding the first 3 questions should be able to answer this question for you.

  1. Comment: “The context for this study is very narrow geographically. Concerns about the effects of land cover conversion are likely global. Concerns about the effects of changing climate and reduction in water availability are also global in nature.  The authors need to improve the global context and importance of this study.”

Response: Thank you for your suggestion. In order to effectively curb climate change and achieve carbon neutrality, the planting area of artificial forests in arid and semi-arid regions worldwide is extensive, which is an important background for this study. Exploring the changes in surface soil moisture content of artificial forests at different stages can provide important data support for understanding the relationship between artificial forest construction and surface soil moisture conditions.

  1. Comment: “The spatial geostatistics analysis does add some degree of novelty to the manuscript. However, it is difficult to match the patterns in Figure 5 to the locations of the various cover types. It would be very helpful to somehow indicate the location of the cover types on that figure. This can be done to some degree by going to Figure 2, however the perspective differs and it is not particularly convenient.”

Response: Thank you for your suggestion. We have added the same boundary line as in Figure 2 to Figure 5 in the hope that this will enhance the readers' understanding.

Specific comments:

  1. Comment: “I don’t think that the title reflects the conclusions correctly. It seems to imply that there are no downward fluctuations in non-forested cover types. However, given seasonal variations all cover types have downward fluctuations as they have upward fluctuations. What seems to have been found is that reforestation leads to a long-term downward trend.”

Response: We express our gratitude for your suggestions. In conjunction with the prior condensation of the main content of the manuscript, we concur with your perspective. We have revised the title of the revised manuscript, please verify its accuracy.

  1. Comment: “I don’t believe that succession is the correct word here. Succession refers to the long-term trend in vegetation composition and structure. As the rest of the abstract does not refer to that I can only assume that the correct word is success.”

Response: Your choice of words is highly precise, and we are willing to accept your suggestions, making modifications in the manuscript accordingly. As a dominant tree species in the local mountainous regions, the success of Picea crassifolia (Qinghai spruce) afforestation primarily depends on the suitability of the selected location. A favorable microenvironment on the ground may sustain the long-term growth of the plantation. Conversely, if the surface soil lacks sufficient moisture content, despite years of artificial irrigation, the plantation will ultimately perish.

  1. Comment: “7 this makes it seem as if there are two entities a native tree and a dominative species. I am assuming they are one and the same?”

Response: Yes, as you have indicated, our expression was indeed inappropriate and potentially mislead. Consequently, we have excised the superfluous descriptions in the revised manuscript.

  1. Comment: “9 The way this is written it seems to suggest the concern about soil moisture is whether it is sufficient to plant trees. However, the thrust of the uncertainty seems to be whether reforestation has had impacts on soil moisture. I suggest the authors revise to make the uncertainty that is the focus of the analysis clearer.”

Response: We appreciate your suggestions and have restructured the main research content of the manuscript accordingly. The modifications have been highlighted in the abstract to elucidate our true research theme more clearly.

  1. Comment: “19 This conclusion is based on an unspecified management objective. Please specify what the objective is for the reader.”

Response: Thank you for your suggestion; the principal conclusions of the study have been reorganized accordingly. Picea crassifolia, serving as a dominant arbor species in local phytocoenoses, can thrive under suitable edaphic conditions. The investigation revealed that natural Picea crassifolia forests typically exhibit a patchy landscape pattern, which is markedly distinct from the homogeneous plantations of artificially cultivated forests. Therefore, exploring the interactive relationship between afforestation and soil moisture content is of profound significance, as it may lead to the identification of the optimal vegetation restoration strategy for the region. The findings of this research could provide theoretical support for regional vegetation restoration initiatives. The modifications have been highlighted in the abstract.

  1. Comment: “39 If the authors already know that planting of this species will lead to these changes, then why conduct the analysis described in the manuscript?”

Response: Following the establishment of artificial forests, a natural interaction inevitably arises between the forest and the surface soil moisture due to the requirements of growth and transpiration processes. However, the precise nature of this interaction remains unclear, as does the extent of the changes that artificial forestation may induce in local surface soil moisture levels. These uncertainties constitute the primary scientific questions that this study seeks to investigate.

In the manuscript, we have revised the original statements and further clarified the correspondence between different successional stages of artificial forests and the vegetation landscape. This refinement aims to provide readers with a more accurate formulation of the scientific questions at hand.

  1. Comment: “54-56 This section seems to compare two entities, but it is not clear what they are.”

Response: Appreciation is extended for the reminder. It has been recognized that the presentation within the original manuscript was unclear. The intention was to convey that, in humid regions, soil moisture content is not the decisive factor in determining vegetation patterns. Amendments have been made in the revised manuscript, for which review is requested.

  1. Comment: “62 controlling factor?”

Response: Thank you for the correction. The concept we intended to convey is that precipitation is the primary driving factor in the evolution of arid regions. We have completed the revision in the modified manuscript.

  1. Comment: “65 I am not sure what “the main zone” means.”

Response: We regret that our previous expression was not sufficiently precise, leading to misunderstandings among our readers. What we intended to convey is that the soil moisture content in the regions where the root systems of vegetation in arid areas are located is of critical importance to the physiological activities and growth of the vegetation. We have restated this in the revised version, hoping that our modifications will clarify our intended message.

  1. Comment: “68 I am not sure the logic behind this statement is true. It seems to leave a number of important factors out.”

Response: Water content, as a core factor in the environmental changes of arid regions, indeed poses a significant limitation on the evolution and development of regional vegetation. Precipitation can affect the growth state of regional vegetation; however, years of stable precipitation conditions will eventually be reflected in the regional soil moisture content. That is to say, in arid regions, the amount of soil moisture is directly linked to the succession pattern of regional vegetation, a point that is substantiated in numerous studies. For instance, in certain literature, it is mentioned that “From 1995 to 2005, the precipitation-controlled shifts between arid and continental zones were mainly concentrated in the Northeast Plain. Due to the decrease in precipitation (Fig. 2b), the continental zone in the Northeast Plain was replaced by arid zones.” [1].

  1. Comment: “80 understanding?”

Response: Thank you, this should be a noun, and we have made the correction in the revised manuscript.

  1. Comment: “92 is evolution needed in this sentence?”

Response: Yes, in this study, we employed the method of "space-for-time" substitution to investigate and analyze the moisture conditions of shallow surface soils in artificial forests with varying planting durations. The findings of this research can offer a reference for understanding the trends in surface soil moisture changes following the establishment of artificial forests in semi-arid regions.

  1. Comment: “102 is the word perfect needed? Perfect is a very high standard to attain.”

Response: Yes, as you have indicated, the diction in our previous manuscript was indeed somewhat exaggerated. We have revised the expression to “relatively ideal”, which may more accurately convey our genuine perspective.

  1. Comment: “121 Would “at the lower elevations” make more sense? Or was it the lowest elevation? As written it is not clear.”

Response: Yes, thank you for your suggestion; we have implemented the revisions in the modified manuscript.

  1. Comment: “125 Is there a more concise way to express this? I take it that the average ratio of forest to grass cover increased 6.2% over the period in question.”

Response: We appreciate your suggestion that relying solely on growth percentages does not adequately address the issue. To this end, we have articulated the changes in forest area between the years 2000 and 2010, with the hope that our revisions will elucidate the alterations in forest cover within the study region. The modified sections have been highlighted for emphasis.

  1. Comment: “149 the formatting of the table makes it difficult to read the headings.”

Response: We appreciate your reminder. In light of the complexity inherent in the plot conditions, our intention is to present all information pertinent to the research content to the readers, thereby facilitating their comprehension of the subsequent material. We trust that the journal's typesetting staff will adeptly manage the formatting of the table headers, and we hope this will not engender confusion among the readers.

  1. Comment: “166 In order is not needed in any sentence. Please remove. Also why was this time period selected other than the precipitation conditions? Does this represent a season low point or average? Or an annual one? It is not clear why.”

Response: We appreciate your reminder and have removed the redundant statements in the revised manuscript. Regarding the setting of survey conditions, we have provided a detailed response in the third question of the introduction. The establishment of weather conditions aims to identify the most optimal survey time window, thereby enabling a more accurate understanding of the surface soil moisture conditions in artificial forests of varying planting ages.

  1. Comment: “179 Please move results to the results section.”

Response: We appreciate your identification of the issue at hand. The statistical results presented in Table 2 have been relocated to section 3.1, under the heading "Results". The Methods section now solely describes the statistical analysis techniques. Following this relocation, the manuscript's layout will require further revision by the publishing editor, for which we are profoundly grateful.

  1. Comment: “185 Again table formatting makes it difficult to read.”

Response: We appreciate your identification of the issue at hand. The statistical results presented in Table 2 have been relocated to section 3.1, under the heading "Results". The Methods section now solely describes the statistical analysis techniques. Following this relocation, the manuscript's layout will require further revision by the publishing editor, for which we are profoundly grateful. In order to enhance the readability of Table 2, we have provided a detailed explanation of the statistical methods used for Table 2 in the Methods section (2.2.3. Statistical Analysis), and have highlighted the text for emphasis.

  1. Comment: “186-224 This section is written more along the lines of a textbook than a methods section. Providing some background to reads is fine, but this has to primarily focus on the actual methods used.”

Response: We appreciate your suggestion and have also sought to reduce the content of this section as much as possible. However, the first two paragraphs pertain to the concept and principles of semivariogram functions, while the subsequent two paragraphs primarily address the key parameters and process settings during the analysis of semivariogram functions within the software. Different combinations of parameters yield distinct implications, which are crucial for readers to comprehend the analysis results and the discussion section. Therefore, after multiple consultations, we have resolved to err on the side of detail rather than risk confusion among readers due to oversimplification. In relation to the parameter setting and result analysis of the semivariogram, a considerable amount of time was devoted to the analysis of the survey data. Consequently, it is our hope that you will comprehend our decision.

  1. Comment: “242 What is the mixed forest? Do the authors mean the natural and planted mixed forest? Please be consistent in use of terms so the reader can understand the results.”

Response: Yes, all the mixed forests mentioned in the manuscript refer to the intermixing of artificial and natural forests. We have also made corresponding revisions in the relevant sections of the document.

  1. Comment: “268 Again table formatting makes it hard to read.”

Response: We sincerely apologize for the suboptimal reading experience provided to our readers due to the excessively large row spacing in the tables of our previous version. In the revised version, we have readjusted the tables with the aim of enhancing their readability.

  1. Comment: “269 This reads like a set of notes. Please revise using complete sentences.”

Response: We sincerely apologize for the suboptimal reading experience provided to our readers due to the excessively large row spacing in the tables of our previous version. In the revised version, we have readjusted the tables with the aim of enhancing their readability.

  1. Comment: “291 Hasn’t this been known for a longer period than since 2014? And in a wider range of locations? Wasn’t this the basis of Clements’ climax?”

Response: It is indeed true that we have had a relatively clear understanding of the relationship between regional average rainfall and vegetation types for quite some time. However, in the face of demands for carbon neutrality prompted by climate change, there exists a need for the planting of artificial forests. This need is constrained by the complexity and long-term nature of the relationship between environmental factors and vegetation growth, and as such, we may not always be able to identify the most optimal vegetative greenery. To address this practical issue, it is imperative that, upon the completion of vegetation construction, we track the growth of artificial vegetation and monitor the soil conditions of the artificial forests, especially the variations in soil moisture content. This will enable us to elucidate the interaction mechanisms between vegetation and its environment, thus providing theoretical support for more efficient greening efforts in the future.

  1. Comment: “301-305 I found this sentence confusing because it combines several ideas that are difficult to sort out as written.”

Response: In our revised manuscript, we have reorganized our language in order to clearly delineate the decrease in soil moisture content observed as artificial forests progress to the mixed forest stage—a phenomenon that warrants our attention. Whether this is a cause of the eventual mortality in artificial forests requires further substantiation. All modifications can be seen in the highlighted sections.

  1. Comment: “326 I am not sure that “in other words” makes sense in this context. That implies there will be an alternative wording, not an elaboration.”

Response: Thank you for your suggestion. The previous statement was indeed not quite rational and the transition was not sufficiently smooth. We have readjusted our linguistic expression to ensure that our intended meaning is comprehensible to all.

  1. Comment: “364 On the other hand and but mean the same thing. Use one or the other, but not both.”

Response: Thank you for your suggestion; we have incorporated the revisions into the modified manuscript.

  1. Comment: “369 Understanding? Also this seems to state that understanding improves understanding. That reads quite oddly.”

Response: We appreciate your reminder and have amended the inaccuracies in the manuscript's expression, with the aspiration that our revisions will enhance the comprehensibility for the readers.

  1. Comment: “371 It should be made clear that spatial and not temporal variability was examined.”

Response: We appreciate your identification of the issue, and we have revised the conclusion section accordingly. The amended segments have been highlighted for ease of reference.

Additionally, we have engaged industry experts who are native English speakers to thoroughly revise the language of the manuscript. It is our hope that our modifications have fully addressed the suggestions and comments put forth by the peer reviewers, while simultaneously enhancing the overall quality of the manuscript. This endeavor aims to provide valuable research references for the pertinent field.

Again, we would like to thank the editor and the two reviewers for your valuable comments. Your comments helped us clarify our writing, and made our manuscript more reader-friendly.

Reviewer 2 Report

Comments and Suggestions for Authors

The MS deals with the effect that the land-cover classes “natural forest, shrubland, grassland, planted forest and mixed forest” has on the water content of surface soils in the semi-arid region of the Qilian Mountains along the Hexi-corridor, NW China. This is a relevant research question because the water yield from the mountains depends decisively on the land-cover as well as erosion caused by heavy rainfall. The results of the study are more or less in-line with results of the Academy of Water Resource Conservation forests of Qilian Mountains in Zhangye worked-out between 2012 and 2014 in the Pailugou catchment adjacent to the study area presented here.

The study is based on a broad data material from 480 sampling point in 10 observation plots. I feel that the potential of this material was by far not used as it would be possible. The MS is not well organized and contains lot of fuzzy interpretation that is not supported by the material and its evaluations. Also formulations and the presentation of graphs and tables are often insufficient (see my detailed comments on both aspects in the text).

My main concern is, that the authors didn´t perform their evaluations on the basis of the data from the individual observation points rather than on plot basis that is integrating over e.g. land-cover classes, slope position etc. Thus the influencing factors on SWC cannot been identified in sufficient accuracy. The only, somehow unsatisfactory approach would be the qualitative interpretation of the spatial distribution of SWC at the kriging maps given in Fig. 5. But also this option is not worked out since e.g. the areas of the different land-cover classes or slope positions are not marked on the plots in Fig. 5 (see my comments in the text).

I would suggest to the authors to review the MS substantially according to my comments and to write explicitly in the discussion and the conclusion that the interpretation of the results is a first, preliminary approach on the basis of merely qualitative interpretation of spatial distribution patterns for SWC from the kriging maps of the 10 observation plot after giving all additional information according to my comments. I want to urge the authors to strictly avoid all over-interpretation of these qualitative evaluation. Perhaps they could add an outlook on future evaluations to the conclusion where an identification of the effects of distinct factors influencing the SWC on the basis of the individual observation points by means of GAMM models will be announced. Such General Additive Mixed Models (GAMM) could provide effect graphs with confidence intervals on influencing factors (fixed effects) like land-cover class, slope position, slope inclination, slope aspect, crown density, etc. with the precondition that these entities are attached to each of the 480 individual sampling points. Additionally the plots could be included in the GAMM models as random factors representing variation that was not captured by the fixed effects. In the face of the high experimental effort already invested I think that such a follow-up evaluation would be worthwhile

Comments on the Quality of English Language

proofreading by a professional native speaker recommended

Author Response

Commends from Reviewer 2:

  1. Comment: “The MS deals with the effect that the land-cover classes “natural forest, shrubland, grassland, planted forest and mixed forest” has on the water content of surface soils in the semi-arid region of the Qilian Mountains along the Hexi-corridor, NW China. This is a relevant research question because the water yield from the mountains depends decisively on the land-cover as well as erosion caused by heavy rainfall. The results of the study are more or less in-line with results of the Academy of Water Resource Conservation forests of Qilian Mountains in Zhangye worked-out between 2012 and 2014 in the Pailugou catchment adjacent to the study area presented here.”

Response: We express our gratitude for the recognition of our research endeavors, which indeed constitute a part of the Qilian Mountains research series. Due to various reasons, these findings have not been published until now. Recently, we have organized the preliminary sampling data and identified several meaningful discoveries. It is our hope that the results of our study will be successfully published in the respective journal. We are prepared to make detailed revisions to the manuscript in accordance with your suggestions. Thank you.

  1. Comment: “The study is based on a broad data material from 480 sampling point in 10 observation plots. I feel that the potential of this material was by far not used as it would be possible. The MS is not well organized and contains lot of fuzzy interpretation that is not supported by the material and its evaluations. Also formulations and the presentation of graphs and tables are often insufficient (see my detailed comments on both aspects in the text).”

Response: Thank you for your suggestions; indeed, we have encountered numerous challenges in our data analysis. Due to the minimal variation in soil moisture content across different vegetation types in the surveyed areas, we have faced considerable obstacles in analyzing the correlation between vegetation types and soil moisture conditions. For instance, effectively presenting the differences in soil moisture content at various successional stages within man-made forests has proven to be a complex task. Of course, the issues present in our current manuscript are also limited by our data analysis capabilities. We are very open to all reasonable suggestions and feedback, and we sincerely hope that our revisions will enhance the quality of our manuscript, thereby facilitating its successful publication.

  1. Comment: “My main concern is, that the authors didn´t perform their evaluations on the basis of the data from the individual observation points rather than on plot basis that is integrating over e.g. land-cover classes, slope position etc. Thus the influencing factors on SWC cannot been identified in sufficient accuracy. The only, somehow unsatisfactory approach would be the qualitative interpretation of the spatial distribution of SWC at the kriging maps given in Fig. 5. But also this option is not worked out since e.g. the areas of the different land-cover classes or slope positions are not marked on the plots in Fig. 5 (see my comments in the text).”

Response: Your opinion is highly valued by us. Globally, research on forest hydrology often yields conclusions that are markedly divergent, and at times, entirely contradictory. This is primarily attributed to the high spatial variability inherent to the subject of study. Due to the considerable spatial variability of forests, research scopes generally need to be sufficiently expansive to circumvent anomalies arising from spatial differences. However, in practical surveys, the research is frequently constrained by the topography and geomorphology of the study area. Often, it is not feasible to locate an ideal research plot that meets criteria such as uniform vegetation type, similar ground conditions, consistent area size, and uniform sampling intervals. Consequently, in our field investigations, the size of the plots we find varies, and the sampling intervals also differ slightly due to micro-topography. This accounts for the somewhat disordered analysis observed in the current manuscript. We hope our explanation will ameliorate your view. Thank you for your suggestions; indeed, we have encountered numerous challenges in our data analysis. Due to the minimal variation in soil moisture content across different vegetation types in the surveyed areas, we have faced considerable obstacles in analyzing the correlation between vegetation types and soil moisture conditions. For instance, effectively presenting the differences in soil moisture content at various successional stages within man-made forests has proven to be a complex task. Of course, the issues present in our current manuscript are also limited by our data analysis capabilities. We are very open to all reasonable suggestions and feedback, and we sincerely hope that our revisions will enhance the quality of our manuscript, thereby facilitating its successful publication.

  1. Comment: “I would suggest to the authors to review the MS substantially according to my comments and to write explicitly in the discussion and the conclusion that the interpretation of the results is a first, preliminary approach on the basis of merely qualitative interpretation of spatial distribution patterns for SWC from the kriging maps of the 10 observation plot after giving all additional information according to my comments. I want to urge the authors to strictly avoid all over-interpretation of these qualitative evaluation. Perhaps they could add an outlook on future evaluations to the conclusion where an identification of the effects of distinct factors influencing the SWC on the basis of the individual observation points by means of GAMM models will be announced. Such General Additive Mixed Models (GAMM) could provide effect graphs with confidence intervals on influencing factors (fixed effects) like land-cover class, slope position, slope inclination, slope aspect, crown density, etc. with the precondition that these entities are attached to each of the 480 individual sampling points. Additionally the plots could be included in the GAMM models as random factors representing variation that was not captured by the fixed effects. In the face of the high experimental effort already invested I think that such a follow-up evaluation would be worthwhile.”

Response: We appreciate your suggestion; the focal point of this study is the intrinsic relationship between soil moisture and the duration of afforestation in artificial forests. The research employs a space-for-time substitution approach to investigate and sample plots of varying afforestation durations, thereby exploring the long-term impact of afforestation in arid and semi-arid regions on the local soil moisture environment. Consequently, the spatial pattern and trend of soil moisture in artificial forests of different ages constitute the core content of our investigation. To provide a more rational explanation for the variations in soil moisture, we have also surveyed other factors in the study plots, although there is typically one set of data per vegetation type.

In your suggestion, you proposed that we utilize the Generalized Additive Mixed Model (GAMM) to perform a non-linear fitting analysis of environmental factors and soil water content. This statistical analysis might indeed reveal the intrinsic correlations between environmental factors and soil water content. However, there is a risk that this additional content may deviate from the main theme of the article. Therefore, we believe it is prudent to maintain the integrity of the original manuscript's core content. In response to the detailed issues you have raised, we will meticulously revise each point.

Specific comments:

  1. Comment: “line 2: what should that itm say? decrease? or minimum peaks?”

Response: Thank you for your suggestion. The term "downward trend" is indeed more appropriate in this context. We have implemented the revision in the modified draft; please review the highlighted section.

  1. Comment: “line 12: are these 2 landcover classes both planted forests? and why they show much lower SWC than natural forests?”

Response: Artificial forests, due to their relatively low planting density, leave a substantial amount of land exposed, resulting in significant soil evaporation. Concurrently, the growth conditions of the herbaceous and shrub layers are suboptimal, as they are overshadowed by the tree canopy. Consequently, both the initial plantings of artificial forests and the mixed forests that have been established for some time exhibit comparatively low soil moisture content. In contrast, natural forests, with their larger canopies, provide effective shading over the ground, substantially reducing surface evaporation. Additionally, the presence of surface mosses and the accumulation of multi-year litter create a rich organic layer that not only retains moisture effectively during rainfall but also mitigates high-temperature evaporation from the soil surface after precipitation events. Therefore, the soil moisture content in natural forest areas is significantly higher than that in regions with artificial forests.

  1. Comment: “line 51: this is a rather trivial finding”

Response: Yes, this is a trivial finding, and this is merely a minor result derived from others' work; hence, we believe it does not present significant issues.

  1. Comment: “line 52: this is an indicator that obviously the local density of vegetation cover, down to the scale of single tree crowns could be a key factor for small -scale vatiation of SWC. It would be worth while to eluciate the relation between local density of vegetation cover (crown density) and SWC at the basis of the individual sampling points - this would be 480 independent observations (see Tab 1)”

Response: Yes, thank you for your suggestion. Our research involved the application of Generalized Additive Mixed Models (GAMM4) to model the relationship between six environmental variables and soil moisture content. The analysis revealed a significant relationship (P<0.001) between soil moisture content and both vegetation type and vegetation height. We will systematically amend this portion in the manuscript.

  1. Comment: “line 58-71: citations don´t follow a reasonable and logic chain of argumentation”

Response: In the process of revising the manuscript, we have restructured this section and employed highlighted backgrounds to display the modified sentences, with the aspiration that our amendments will articulate our authentic perspective with clarity.

  1. Comment: “line 91: theoretically sound working hypotheses, but in detail they don´t fit to the data and the boundary conditions of the study”

Response: Yes, in accordance with the revised manuscript content, we have also modified the primary scientific questions of interest in this study.

11 Comment: “line 92-93: I don´t see the opportunity to differentiate "early stages of plantations" in the sampling plots No 1-10 in the sense of a time series that would be the precondition of such evaluation.”

Response: In this study, the surveyed plantations predominantly have an age of less than 50 years, with mature forests being approximately 80 years old. Previously, we defined the initial phase of plantation as being within 50 years, a terminology that may pose reading difficulties for the reader. Consequently, we have revised our terminology and highlighted these changes within the text, in hopes that our modifications accurately convey our intentions.

12 Comment: “line 94-95: I don´t see such an approach in this study.”

Response: We sincerely apologize, as the content corresponding to this scientific inquiry was inadvertently deleted during the revision process. The question itself has also been modified, and in accordance with the recommendations of the reviewers, we have adopted the Generalized Additive Mixed Model (GAMM) to refine our scientific query based on the simulation results.

13 Comment: “line 106: mention here the FAO-classes that are given on the next page”

Response: Thank you for your reminder, we have already made annotations in the revised draft.

14 Comment: “line 109: should be organized as explosion plot combining a map that show the position of the study area in China with that detailed map of the study plots”

Response: Modifications have been made to the map of the study area in the revised manuscript. Within the map of China's administrative divisions, the study area is demarcated by a red rectangle. However, due to the limited extent of the survey area, it is almost represented as a red dot on the administrative division map. The updated map provides a clear visualization of the study area's location within China.

15 Comment: “line 147-148: the red lines indicating the boundaries of the plots are not very informative since the don´t fit well to the horizontal and vertical dimensions given in Tab. 1”

Response: Yes. In Table 1, our focus was on the statistical analysis of the study area based on sample plots. However, in the subsequent analysis, our primary focus shifted to the statistical evaluation of different vegetation types. Therefore, it is imperative to conduct a stratified statistical analysis of composite sample plots according to vegetation types, thereby facilitating the investigation of the relationship between various landscape types and soil moisture content within this study. In Figure 2, the boundaries of individual sample plots appear to be inclined. This is partly due to the constraints imposed by the positions available for photography. An oblique shooting angle can cause the standard rectangular plots to appear skewed, although our survey plots were maintained as standard rectangles as far as possible. Concurrently, the actual survey area is influenced by the terrain, and to circumvent areas that are inaccessible, some sample plots are compromised to be non-perpendicular or non-horizontal.

16 Comment: “line 149: is that the number of sampling points in the individual plots and thus the basis for the kriging interpolation in Fig. 5?”

Response: Yes. The sample size collected in this table constitutes the dataset for our spatial variability analysis in Fig. 5.

17 Comment: “line 149: It would be helpful to add information on slope position of the plots (e.g. upper, middle downward slope position)”

Response: During our investigation, the sample plots were selected to be as consistent as possible in terms of slope position and aspect. Therefore, in the analysis of our study, we analyzed different slope positions using the same aspect without conducting separate measurements for the composite slope aspects of the sample plots. We hope that our experimental design will garner your understanding and support.

18 Comment: “line151-152: How this determination was tried to do?”

Response: We aim to eliminate the spatial variability in soil moisture caused by lateral flow associated with different slope positions through the design of horizontal transects. This approach will enable a more objective statistical analysis of the spatial heterogeneity of soil moisture corresponding to different vegetation types.

19 Comment: “line153-154: obviously that was dane with the kriging maps in Fig 5 but there also plots 6, 7 and 8 were presented.”

Response: We appreciate your prompt. The previous articulation of our position lacked sufficient clarity. Consequently, we have amended this section in the revised manuscript with the aim of enhancing comprehension for the reader.

20 Comment: “line155-156: a sketch with the transects and the sampoling points would be helpful”

Response: We sincerely apologize for the inconvenience. Previously, we had created a draft illustrating the spatial distribution of sample points. However, due to the high density of these points and the desire to clearly display all sampling sites, the resultant visual representation suffered from information overload, which significantly diminished its effectiveness. Consequently, we decided to forgo the inclusion of the sampling point distribution in the design. We hope for your understanding and support in this matter.

21 Comment: “line164: what is "theoretical" here”

Response: Standard value or reference value. This value is primarily utilized to calibrate our Time-Domain Reflectometry (TDR) observational results, thereby enhancing the accuracy of our instrument measurements.

22 Comment: “line 255: or greater variability in crown cover and subsequently in throughfall intensity.”

Response: We appreciate your suggestions and have accordingly revised the sentences within the manuscript to enhance the precision of expression.

23 Comment: “line 284: Entity Vol%?”

Response: Yes, the image indeed represents the volumetric water content of the soil.

24 Comment: “line 285: Is there the spatial distribution of rel. SWC presented or the CV (coefficient of variance) of SWC?”

Response: Yes, the diagram depicts the spatial variation of the coefficient of variation for soil moisture content across different sample locations.

25 Comment: “line 286: I would be nice if the boundaries between landcover classes (yellow lines in Fig 2) would be given here as well”

Response: We appreciate your suggestion. Similar to the recommendation provided by another peer reviewer, we have incorporated demarcation lines for different vegetation types into the figures in the revised manuscript.

26 Comment: “line 295-297: true, but unsharp”

Response: This passage merely presents a rudimentary inference regarding the alteration in soil moisture content subsequent to the establishment of artificial forests. The focal point of this study is to ascertain the potential perils that arid regions may confront in response to such ecological transformations, as well as to distill the lessons and insights that can be gleaned from these changes. In the revised manuscript, appropriate modifications have been made to this section, which are highlighted to ensure that our contemplations are expressed more comprehensively.

27 Comment: “line 298-299: was about 72% lower as compared to...”

Response: Thank you, we have already made the modifications.

28 Comment: “line 300-301: This sentence is not clear”

Response: Thank you for the reminder. Indeed, the expression in the corresponding section of the original manuscript contained numerous improprieties. We have thoroughly revised this section in the amended manuscript, as detailed in the highlighted portions.

29 Comment: “line 301-305: also this is rather confusing”

Response: Indeed, the expression in the corresponding section of the original manuscript contained numerous improprieties. We have thoroughly revised this section in the amended manuscript, as detailed in the highlighted portions.

30 Comment: “line 321-322: I see that tendency only in plots 3 and 4. But this is only a visual guesss and cannot be quantified like it would be possible e.g. with effect graphs and their confidence intervals of GAMM models (see general comment in my review)”

Response: We appreciate the reminder. In accordance with your suggestion, we have conducted a multifactorial analysis modeling using Generalized Additive Models (GAMM) for SWC, slope gradient, aspect, forest age, tree height, and stand density. The results of the modeling will be elaborated upon in detail throughout the manuscript, with corresponding content added to each section. We hope that our revisions will render the interpretation of the issue more objective and scientific.

31 Comment: “line 321-322: I see that tendency only in plots 3 and 4. But this is only a visual guesss and cannot be quantified like it would be possible e.g. with effect graphs and their confidence intervals of GAMM models (see general comment in my review)”

Response: Thank you for the suggestion; we have completed the modeling and analysis based on Generalized Additive Mixed Models (GAMM).

32 Comment: “line 341-344: Autocorrelation distance was calculated at plot basis. Since several plots (No 3, 6, 7, 8, 9) contain several landcover classes, the autocorrelation distance is an information integrating over these landcover classes. This it cannot be interprested like in this part of the text.”

Response: “Range (A, also called effective range) is the separation distance over which sample locations are autocorrelated, i.e. over which there is calculated from the range parameter A0 (anisotropic models) or A1 and A2 [1]”. “the range parameter may be different from the effective range, which should be used to compare ranges among models”, P68: “In GS+ the Range (A) is calculated from A0 as described in the formulas for the different models”.

Indeed, as you have indicated, the value (A) cannot be simply employed to elucidate the effective range of semivariogram analysis for a specific vegetation type within a mixed plot. Nevertheless, it can still serve as an indicator for analyzing the semivariogram effective distances between different vegetation types and mixed plots. Should there be any issues with our understanding, we kindly request your correction.

33 Comment: “line 369: cannot be clearly identified with kriging and variogram analyses on plot basis”

Response: We appreciate your suggestion and have also recognized that the analysis of the primary influencing factors in the original manuscript largely stems from our own reasoning and conjecture, which is insufficient to persuade the readers. Heeding your advice, we have employed the Generalized Additive Mixed Model (GAMM) to conduct a more objective analysis and evaluation of our survey data.

Additionally, we have engaged industry experts who are native English speakers to thoroughly revise the language of the manuscript. It is our hope that our modifications have fully addressed the suggestions and comments put forth by the peer reviewers, while simultaneously enhancing the overall quality of the manuscript. This endeavor aims to provide valuable research references for the pertinent field.

Again, we would like to thank the editor and the two reviewers for your valuable comments. Your comments helped us clarify our writing, and made our manuscript more reader-friendly.

Round 2

Reviewer 1 Report

Comments and Suggestions for Authors

Thank you for your revisions. 

Author Response

Commends from Reviewer 1:

Common reviews: “Thank you for your revisions.”

Response: We express our gratitude for your suggestions and also appreciate your affirmation of our revised manuscript.

Reviewer 2 Report

Comments and Suggestions for Authors

I appreciate that the authors tried to follow the suggestions in my 1st report to some extent. The MS was enhanced in severeal part substantially. E.g. the indication of vegetation types is now much more consistenet in the figures 2 and 5. Also several parts in the text were re-written in more clarity. Also lot of the responses to my commets were adequate and satisfactory. Nevertheless some aspects are still inconsistent and confusing. This must be enhanced before publication.

With the following respond on my comment I cannot agree:

"17 Comment: “line 149: It would be helpful to add information on slope position of the plots (e.g. upper, middle downward slope position)”
Response: During our investigation, the sample plots were selected to be as consistent as possible in terms of slope position and aspect. Therefore, in the analysis of our study, we analyzed different slope positions using the same aspect without conducting separate measurements for the composite slope aspects of the sample plots. We hope that our experimental design will garner your understanding and support." Beacause the majority of plots (1,2,3,4,5,9,10) contain a pronounced differentiation in slope positions since they reach from ridges to the foot-slope, I am sure that slope positition will cause a sustantial variation of SWC within these plots along the gradient of slope positions. If this variation is not considered explicitly in the evaluations, slope positition cannot be judged adequately in its effect on SWC.

Generally I feel that the authors didn´t follow a stringent evaluation stratey to answer the research questions addressed to in the title and the working hypotheses. In my perception the target variable is in any case SWC and potential influencing variables are vegetation types grass, shrub, forest, forest age, tree density in forests, slope inclination, slope position (I suppose that because of lateral flow foot-slpe positions are more wet than middle or upper slope positions), slope aspect. Nothing but the effect ofte these influencing variables on the target variable SWC must be revealed by the different evaluation steps. Fig 3 gives a more or less qualitative overview on the mean SWC and its variability at plot respectively sub-plot level (areas with diferent vegetation types within plots). The same is given with focus on the influencing variabole vegetation type in Fig. 4. In both cases the variation around the mean values is substantially high which results from the fact that these values derived at the basis of plots or sub-plots contains substantial intrinsic variation caused e.g. by slope positions, differing vegetation density etc. Also Kriging maps can resolve this intrinsic variability only partially because they are interpolated, and that only with the precondition that in Fig 5 the Kriging interpolation would be applied for SWC and not for CV (SWC). The only way th reduce and to explain this unexplained variability is to apply an multiple reggrssion approach (e.g. OLS) or as I suggested a GAMM approach (that has the advantage to provide robust, non-parametric and smooth link-functions) at the level of the individual observations. Forcing precondition for the GAMM modelling is that all interesting influencing variables are attached to each individual observation. I cannot see that this was performed in the GAMM model added to the revised version of the MS. In any case the methodical details of the GAMM modelling must be explained in much more detail and also the explanation of its coefficients in the results chapter.

Please consider the detailed comments that I inserted directly to the text.

Comments on the Quality of English Language

Partly not easy to read

Author Response

Commons from Reviewer 2:

  1. Common reviews: “I appreciate that the authors tried to follow the suggestions in my 1st report to some extent. The MS was enhanced in several part substantially. E.g. the indication of vegetation types is now much more consistent in the figures 2 and 5. Also several parts in the text were re-written in more clarity. Also lot of the responses to my comments were adequate and satisfactory. Nevertheless some aspects are still inconsistent and confusing. This must be enhanced before publication.

With the following respond on my comment I cannot agree:

"17 Comment: “line 149: It would be helpful to add information on slope position of the plots (e.g. upper, middle downward slope position)”
Response: During our investigation, the sample plots were selected to be as consistent as possible in terms of slope position and aspect. Therefore, in the analysis of our study, we analyzed different slope positions using the same aspect without conducting separate measurements for the composite slope aspects of the sample plots. We hope that our experimental design will garner your understanding and support." Beacause the majority of plots (1,2,3,4,5,9,10) contain a pronounced differentiation in slope positions since they reach from ridges to the foot-slope, I am sure that slope positition will cause a sustantial variation of SWC within these plots along the gradient of slope positions. If this variation is not considered explicitly in the evaluations, slope positition cannot be judged adequately in its effect on SWC.

Generally I feel that the authors didn´t follow a stringent evaluation stratey to answer the research questions addressed to in the title and the working hypotheses. In my perception the target variable is in any case SWC and potential influencing variables are vegetation types grass, shrub, forest, forest age, tree density in forests, slope inclination, slope position (I suppose that because of lateral flow foot-slope positions are more wet than middle or upper slope positions), slope aspect. Nothing but the effect of these influencing variables on the target variable SWC must be revealed by the different evaluation steps. Fig 3 gives a more or less qualitative overview on the mean SWC and its variability at plot respectively sub-plot level (areas with diferent vegetation types within plots). The same is given with focus on the influencing variabole vegetation type in Fig. 4. In both cases the variation around the mean values is substantially high which results from the fact that these values derived at the basis of plots or sub-plots contains substantial intrinsic variation caused e.g. by slope positions, differing vegetation density etc. Also Kriging maps can resolve this intrinsic variability only partially because they are interpolated, and that only with the precondition that in Fig 5 the Kriging interpolation would be applied for SWC and not for CV (SWC). The only way the reduce and to explain this unexplained variability is to apply an multiple reggrssion approach (e.g. OLS) or as I suggested a GAMM approach (that has the advantage to provide robust, non-parametric and smooth link-functions) at the level of the individual observations. Forcing precondition for the GAMM modelling is that all interesting influencing variables are attached to each individual observation. I cannot see that this was performed in the GAMM model added to the revised version of the MS. In any case the methodical details of the GAMM modelling must be explained in much more detail and also the explanation of its coefficients in the results chapter.

Please consider the detailed comments that I inserted directly to the text.”

Response: First and foremost, we extend our sincere gratitude for your patient assistance in resolving the issue at hand. In the revised manuscript, we indeed conducted a Generalized Additive Mixed Model (GAMM) analysis on the relationships between most of the latent variables (including vegetation type, forest age, slope gradient, aspect, vegetation height, vegetation density, and survey plot area) and the target variable (Soil water content). We have supplemented the sections 193-203 of the revised manuscript with detailed descriptions of the data types for all variables involved, and the previous analysis results have indeed revealed certain definitive variable relationships. However, as you have pointed out, we did not previously include the slope position factor. In this revision, we will incorporate this element and reanalyze the multifactorial relationships using GAMM. We will elucidate the results in the "Results" and "Discussion" sections, and we hope that our modifications will thoroughly address this critical issue.

Special comments:

  1. Comment: “Line96-97: that is not informative. please write here explicitly what the variables are that you suppose to influence SWC”

Response: We have refined this section in the revised manuscript with the aim of enhancing the reader's comprehension.

  1. Comment: “Table 1: format please in such a way that the headers are easily to read”

Response: We appreciate your affirmation of our modifications.

  1. Comment: “Line 193-203: I appriciate very much that you followed my suggestion to set-up a GAMM model for identification of relevant predictors for SWC. But I miss here the important technical details of the modelling approach:

-> what were the criteria of parameter selection to ensure a parsimonious fit of the model and to avoid over-fitting?

->  what was the formula of the final model?

-> I don´t understand why the plot area was introduced as random effect since the sice of the plots is a scalar. I suggested with my 1st review to introduce plot number as random effect and thus capturing not measured effects of the plot situations”

Response: All observed environmental factors potentially related to soil moisture content variation were included in the analysis. The primary environmental factors encompassed plot number (ID), vegetation type (VEG), forest age (AGE), slope position (SP), slope gradient (SLP), slope aspect (SLPA), vegetation height (HIG), vegetation density (DEN), survey plot area (ARE), and sample size (NUM), among others. To eliminate the phenomenon of multicollinearity among factors, a Variance Inflation Factor (VIF) test was conducted on all numerical factors, resulting in the removal of the highly correlated factor, vegetation height (HIG). Furthermore, to avoid statistical dimensionality issues caused by varying measurement standards of the observational factors, a standardization transformation was applied to all numerical factors, thereby ensuring the robust construction of the Generalized Additive Mixed Models (GAMM).

        Analysis revealed that within the Generalized Additive Mixed Models (GAMM), the family was identified as Gaussian with an R-squared value of 0.32. The log-likelihood value using Restricted Maximum Likelihood (REML) for the linear mixed-effects model (lmer) was -441.93, and the scale estimate stood at 0.02. The sample size for this study was 480. The final model employed was the Linear Mixed-Effects Models (lmer).

  1. Comment: “Line 282: ....shrublands and grasslands, that in all probability is caused by the higher heterogneneity in vegetation cover between tree crowns and gaps in forests.”

Response: We appreciate your explanation; the perspective will be supplemented in the discussion section.

  1. Comment: “Line 308-316: The authors confirmed in their response letter to reviewer (see comment 24), that the Kriging maps indeed show the local variability of the coefficient of variance of SWC. This is in my peception a more or less confusing item with no substantial relation to the title and the aim of the study that is simply the soil water content (see the title and working hypothesis b. in the introduction). In that context it would be adequate to provide kriging maps for the spatial distribution of SWC and not for CV of SWC.”

Response: Our initial intention was to reflect the spatial pattern of soil moisture by examining the spatial variability of soil water content, an approach that we acknowledge may have certain deviations. Following your suggestion, we have applied the Kriging method to perform spatial interpolation on various sample sites, the results of which can be seen in the new figure. We will also reanalyze the results and discussion in accordance with the spatial pattern map of soil moisture.

  1. Comment: “Line 317-321: I expect here a much more detailed interpretation of the parameter- relations in the GAMM model. superficially I see here some contradiction between the relations in figure 4 where natural forest has the highest values of SWC and planted or mixed forests significantly lower SWC. But in the GAMM model the contribution of the factor "natural forest" on SWC seems to be substantially lower as that of the two other forest types.

You should clarify this!

I ask myself why the factor grassland and shrubland are not included in the GAMM modelling since all vegetation types must be included if their contribution to the variation of SWC should be properly identified.”

Response: We appreciate your suggestion. In accordance with your advice, we have re-conducted the Generalized Additive Mixed Modeling (GAMM), ranging from the analysis of collinearity among observational indicators to the standardization of all numerical variables involved in the analysis, as well as the configuration of random effects within the model. Consequently, we have obtained more scientifically robust analytical results. All modifications are documented in the updated Table 4 and the corresponding text.

  1. Comment: “Line 336-342: I don´t understand this sentence”

Response: In this study, the mean soil moisture content was found to be lower than the results obtained by Zhu Xi et al., which may be attributed to differing antecedent rainfall conditions during the survey period. It is posited that shorter intervals between precipitation events prior to the survey could lead to an overestimation of the average soil moisture content.

Mixed forests generally originate from the phenomenon where, after a portion of a plantation forest that has been cultivated for approximately a decade dies, reforestation is conducted through supplementary planting. Consequently, the lower soil moisture content in mixed forests may indicate an objective phenomenon of reduced water availability that could have occurred during the plantation phase.

We have re-adjusted our explanation pertaining to this section, with the aspiration that our endeavors will elucidate the issue at hand.

  1. Comment: “Line 348-349: this statement is trivial and I don´t understand it, since I think that Tab. 4 shows the parameters of the effect of the different predictiors on SWC and not the correlation among the predictors.”

Response: Thank you for your suggestion; we have reformulated that section accordingly.

  1. Comment: “Line 350-355: the authors must emphasize and interpret the sign of the coefficients in Tab 4”

Response: Thank you so much, we have reformulated that section accordingly. In accordance with your revision suggestions, we have conducted a comprehensive reevaluation of the manuscript based on the new analytical outcomes. It is our aspiration that our endeavors will receive your approval. Naturally, should there remain any isolated issues within the revised draft, we hope to have the opportunity to benefit from your further guidance.

Additionally, we have engaged industry experts who are native English speakers to thoroughly revise the language of the manuscript. It is our hope that our modifications have fully addressed the suggestions and comments put forth by the peer reviewers, while simultaneously enhancing the overall quality of the manuscript. This endeavor aims to provide valuable research references for the pertinent field.

Again, we would like to thank the editor and the two reviewers for your valuable comments. Your comments helped us clarify our writing, and made our manuscript more reader-friendly.
